# DASH: Warm-Starting Neural Network Training in Stationary Settings without Loss of Plasticity

**Baekrok Shin** [*]   **Junsoo Oh** [*]   **Hanseul Cho**   **Chulhee Yun**
Kim Jaechul Graduate School of AI, KAIST
{br.shin, junsoo.oh, jhs4015, chulhee.yun}@kaist.ac.kr

## Abstract

Warm-starting neural network training by initializing networks with previously learned weights is appealing, as practical neural networks are often deployed under a continuous influx of new data. However, it often leads to *loss of plasticity*, where the network loses its ability to learn new information, resulting in worse generalization than training from scratch. This occurs even under stationary data distributions, and its underlying mechanism is poorly understood. We develop a framework emulating real-world neural network training and identify noise memorization as the primary cause of plasticity loss when warm-starting on stationary data. Motivated by this, we propose **Direction-Aware SHrinking (DASH)**, a method aiming to mitigate plasticity loss by selectively forgetting memorized noise while preserving learned features. We validate our approach on vision tasks, demonstrating improvements in test accuracy and training efficiency.[1] [2]

## 1 Introduction

When training a neural network on a gradually changing dataset, the model tends to lose its *plasticity*, which refers to the model's ability to adapt to new information (Dohare et al., 2021; Lyle et al., 2023b; Nikishin et al., 2022). This phenomenon is particularly relevant in scenarios with non-stationary data distributions, such as reinforcement learning (Igl et al., 2020; Nikishin et al., 2022) and continual learning (Chen et al., 2023; Kumar et al., 2023; Wu et al., 2021). While requiring to overwrite outdated knowledge as the environment changes, models overfitted to previously encountered environments often struggle to cumulate new information, which in turn leads to reduced generalization performance (Lyle et al., 2023b). Under this viewpoint, various efforts have been made to mitigate the loss of plasticity, such as resetting layers (Nikishin et al., 2022), regularizing weights (Kumar et al., 2023), and modifying architectures (Lee et al., 2023; Lyle et al., 2023a; Nikishin et al., 2023).

Perhaps surprisingly, a similar phenomenon occurs in supervised learning settings, even where new data points sampled from a stationary data distribution are added to the dataset during training. It is counterintuitive, as one would expect advantages in both generalization performance and computational efficiency when we *warm-start* from a model pre-trained on data points of the same distribution. For a particular example, when a model is pre-trained using a portion of a dataset and then we resume the training with the whole dataset, the generalization performance is often worse than a model trained from scratch (i.e., *cold-start*), despite achieving similar training accuracy (Ash and Adams, 2020; Berariu et al., 2021; Igl et al., 2020). Liu et al. (2020) report a similar observation: training neural networks with random labels leads to a spurious local minimum which is challenging to escape from, even when retraining with a correctly labeled dataset. Interestingly, Igl et al. (2020) found that pre-training with random labels followed by the corrected dataset yields

---

[*]Authors contributed equally to this paper.
[1]The NVIDIA GPU implementation can be found on `github.com/baekrok/DASH`.
[2]For the Intel Gaudi implementation, visit: `github.com/NAVER-INTEL-Co-Lab/gaudi-dash`.

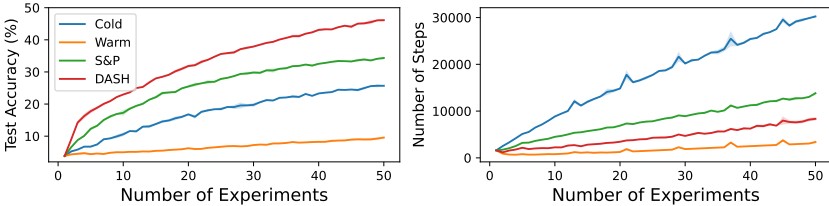

Figure 1: Performance comparison of various methods on Tiny-ImageNet using ResNet-18. The same hyperparameters are used across all methods. The dataset is divided into 50 chunks, with a constant number of data points added to the training dataset in each experiment (x-axis), reaching the full dataset at the 50th experiment. Models are trained until achieving 99.9% train accuracy before proceeding to the next experiment; the plot on the right reports the number of update steps executed in each experiment. Results are averaged over three random seeds. "Cold" refers to cold-starting and "Warm" refers to warm-starting. The Shrink & Perturb (S&P) method involves shrinking the model weights by a constant factor and adding noise (Ash and Adams, 2020). Notably, DASH, our proposed method, achieves better generalization performance compared to both training from scratch and S&P, while requiring fewer steps to converge.

better generalization performance than pre-training with a small portion of the (correctly labeled) dataset and then training with the full, unaltered dataset. It is striking that warm-starting leads to such a severe loss of performance, even worse than that of a cold-started model or a model re-trained from parameters pre-trained with random labels, despite the stationarity of the data distribution.

These counterintuitive results prompt us to investigate the underlying reasons for them. While some studies have attempted to explain the loss of plasticity in deep neural networks (DNNs) under non-stationarity (Lewandowski et al., 2023; Lyle et al., 2023b; Sokar et al., 2023), their empirical explanations rely on various factors, such as model architecture, datasets, and other variables, making it difficult to generalize the findings (Lewandowski et al., 2023; Lyle et al., 2023a). Moreover, there is limited research that explores why warm-starting is problematic in stationary settings, highlighting the lack of a fundamental understanding of the loss of plasticity phenomenon in both stationary and non-stationary data distributions.

## 1.1 Our Contributions

In this work, we aim to explain why warm-starting leads to worse generalization compared to cold-starting, focusing on the stationary case. We propose an abstract framework that combines the popular feature learning framework initiated by Allen-Zhu and Li (2020) with a recent approach by Jiang et al. (2024) that studies feature learning in a combinatorial and abstract manner. Our analysis suggests that warm-starting leads to overfitting by memorizing noise present in the newly introduced data rather than learning new features.

Inspired by this finding, we propose Direction-Aware SHrinking (DASH), which aims to encourage the model to forget memorized noise without affecting previously learned features. This enables the model to learn features that cannot be acquired through warm-starting alone, enhancing the model's generalization ability. We validate DASH using an expanding dataset setting, similar to the approach in Ash and Adams (2020), employing various models, datasets, and optimizers. As an example, Figure 1 shows promising results in terms of both test accuracy and training time.

## 1.2 Related Works

**Loss of Plasticity.** Research has aimed to understand and mitigate loss of plasticity in non-stationary data distributions. Lewandowski et al. (2023) explain that loss of plasticity co-occurs with a reduction in the Hessian rank of the training objective, while Sokar et al. (2023) attribute it to an increasing number of inactive neurons during training. Lyle et al. (2023b) find that changes in the loss landscape curvature caused by non-stationarity lead to loss of plasticity. Methods addressing this issue in non-stationary settings include recycling dormant neurons (Sokar et al., 2023), regularizing weights towards initial values (Kumar et al., 2023), and combining techniques (Lee et al., 2023) like layer normalization (Ba et al., 2016), Sharpness-Aware Minimization (SAM) (Foret et al., 2020), resetting layers (Nikishin et al., 2022), and Concatenated ReLU activation (Shang et al., 2016).

However, these explanations and methods diverge from the behavior observed in stationary data distributions. Techniques aimed at mitigating loss of plasticity under non-stationarity are ineffective under stationary distributions, as shown in Appendix C.1, in line with the observations in Lee et al. (2023). While some works study the warm-starting problem in stationary settings, they rely on empirical observations without theoretical analysis (Achille et al., 2018; Ash and Adams, 2020; Berariu et al., 2021). The most relevant work by Ash and Adams (2020) introduces the Shrink & Perturb (S&P) method, which mitigates the loss of plasticity in stationary settings to some extent by shrinking all weight vectors by a constant factor and adding noise. However, they do not explain why this phenomenon occurs or why S&P is effective. We develop a theoretical framework explaining why warm-starting suffers even under stationary distribution. Based on findings, we propose a method that shrinks the weight vector in a direction-aware manner to maintain properly learned features.

**Feature Learning in Neural Networks.** Recent studies have investigated how training methods and network architectures influence generalization performance, focusing on data distributions with label-dependent features and label-independent noise (Allen-Zhu and Li, 2020; Cao et al., 2022; Deng et al., 2023; Jelassi and Li, 2022; Oh and Yun, 2024; Zou et al., 2023). In particular, Shen et al. (2022) examine a data distribution consisting of varying frequencies of features and large strengths of noise, emphasizing the significance of feature frequencies in learning dynamics. Jiang et al. (2024) propose a novel feature learning framework based on their observations in real-world scenarios, which also involves features with different frequencies but considers the learning process as a discrete sampling process. Our framework extends these ideas by incorporating features with varying frequencies, noise components, and the discrete learning process while introducing a more intricate learning process capturing the key aspects of feature learning dynamics in expanding datasets.

## 2 A Framework of Feature Learning

### 2.1 Motivation and Intuition

We present the motivation and intuition behind our framework before delving into the formal description. Our framework captures key characteristics of image data, where the input includes both label-relevant information (referred to as *features*, e.g., cat faces in cat images) and label-irrelevant information (referred to as *noise*, e.g., grass in cat images). A key intuition is that minimizing training loss involves two strategies: *learning features* and *memorizing noise*. This framework builds on insights from Shen et al. (2022) and integrates them into a discrete learning framework. We provide more detailed intuition on our framework, including the training process, in Appendix A.

Shen et al. (2022) consider a neural network trained on data with features of different frequencies and noise components stronger than the features. The negative gradient of the loss for each single data point aligns more with the noise than the features due to the larger scale of noise, making the model more likely to memorize noise rather than learn features. However, an identical feature appears in many data points, while noise appears only once and does not overlap across data points. Thus, if a feature appears at a sufficiently high frequency in the dataset, the model can learn the feature. Thus, the learning of features or noise depends on the frequency of features and the strength of the noise.

Inspired by Shen et al. (2022), we propose a novel discrete feature learning framework. This section introduces a framework describing a single experiment, while Section 3 analyzes expanding dataset scenarios. As our focus is on gradually expanding datasets, carrying out the (S)GD analysis over many experiments as in Shen et al. (2022) is highly challenging. Instead, we adopt a discrete learning process similar to Jiang et al. (2024) but propose a more intricate process reflecting key ideas from Shen et al. (2022). In doing so, we generalize the concept of plasticity loss and analyze it without assuming any particular hypothesis class for a more comprehensive understanding, whereas existing works are limited to specific architectures.

### 2.2 Training Process

We consider a classification problem with $C$ classes, and data are represented as $(\boldsymbol{x}, y) \in \mathcal{X} \times [C]$, where $\mathcal{X}$ denotes the input space. A data point is associated with a combination of class-dependent features $\mathcal{V}(\boldsymbol{x}) \subset \mathcal{S}_y$ where $\mathcal{S}_c = \{v_{c,1}, v_{c,2}, \ldots, v_{c,K}\}$ is the set of all features for each class $c \in [C]$. Also, every data point contains point-specific noise which is class-independent.

The model $f : \mathcal{X} \to [C]$ sequentially learns features based on their frequency. The training process is described by the set of learned features $\mathcal{L} \subset \mathcal{S} \triangleq \bigcup_{c \in [C]} \mathcal{S}_c$ and the set of data points with non-zero

gradients $\mathcal{N} \subset \mathcal{T}$, where $\mathcal{T} = \{(\boldsymbol{x}_i, y_i)\}_{i \in [m]}$ denotes a training set. The set $\mathcal{N}$, representing the data points with non-zero gradients, will be defined below. The frequency of a feature $v$ in data points belonging to $\mathcal{N}$ is denoted by

$$g(v; \mathcal{T}, \mathcal{N}) = \frac{1}{|\mathcal{T}|} \sum_{(\boldsymbol{x}, y) \in \mathcal{N}} \mathbb{1}(v \in \mathcal{V}(\boldsymbol{x})),$$

where $\mathbb{1}(\cdot)$ is the indicator function, which equals 1 if the condition inside the parentheses is true and 0 otherwise. At each step of training, if $\mathcal{L}$ and $\mathcal{N}$ are given, the model chooses the most frequent feature among the features not yet learned, i.e., arbitrarily choose $v \in \arg\max_{u \in \mathcal{S} \setminus \mathcal{L}} g(u; \mathcal{T}, \mathcal{N})$.

The model decides whether to learn a selected feature $v$ by comparing its signal strength, represented by $|\mathcal{T}| \cdot g(v; \mathcal{T}, \mathcal{N})$, with the signal strength of noise, given by $\gamma$, which reflects the key ideas of Shen et al. (2022). If the frequency of the selected feature $v$ is no less than the threshold $\gamma/|\mathcal{T}|$, i.e., $g(v; \mathcal{T}, \mathcal{N}) \geq \gamma/|\mathcal{T}|$, the model learns $v$ and adds it to its set of learned features $\mathcal{L}$. The feature learning process continues until the model reaches a point where the selected feature $v$ has $g(v; \mathcal{T}, \mathcal{N}) < \gamma/|\mathcal{T}|$, indicating that the signal strength of every remaining feature is weaker than that of noise. At this point, the feature learning process ends.

We consider a data point $\boldsymbol{x}$ to be *well-classified* if the model $f$ has learned at least $\tau$ features from $\mathcal{V}(\boldsymbol{x})$, i.e., $|\mathcal{L} \cap \mathcal{V}(\boldsymbol{x})| \geq \tau$, where $\tau < K$. In this case, we consider $\boldsymbol{x}$ to have a zero gradient, meaning it cannot further contribute to the learning process. Throughout the feature learning process, the set $\mathcal{N}$ of data points with non-zero gradients is dynamically updated as new features are learned. At each step, when the model successfully learns a new feature, we update $\mathcal{N}$ by removing the data points that satisfy $|\mathcal{L} \cap \mathcal{V}(\boldsymbol{x})| \geq \tau$, as they become well-classified due to the newly learned feature.

If the feature learning process ends and the model has learned as many features as it can, the remaining data points that have non-zero gradients will be *memorized* by fitting the random noise present in them and will be considered to have zero gradients. This step concludes the training process. Pictorial illustration can be found in Appendix A and a detailed algorithm of the learning process can be found in Algorithm 2 in Appendix E.

### 2.3 Discussion on Training Process

In our framework, the model selects features based on their frequency in the set of unclassified data points $\mathcal{N}$. The intuition behind this approach is that features appearing more frequently in the set of data points will have larger gradients, leading to larger updates, and we treat $g(v; \mathcal{T}, \mathcal{N})$ as a proxy of the gradient for a particular feature $v$. As a result, the model prioritizes sequentially learning these high-frequency features. However, if the frequency $g(v; \mathcal{T}, \mathcal{N})$ of a particular feature $v$ is not sufficiently large, such that the total occurrence of $v$ is less than the strength of the noise, i.e., $|\mathcal{T}| \cdot g(v; \mathcal{T}, \mathcal{N}) < \gamma$, the model will struggle to learn that feature. Consequently, the model will prioritize learning the noise over the informative features. When this situation arises, the learning procedure becomes sub-optimal because the model fails to capture the true underlying features of the data and instead memorizes the noise to achieve high training accuracy.

The threshold $\tau$ determines when a data point is considered well-classified and acts as a proxy for the dataset's complexity. A higher $\tau$ requires the model to learn more features for correct predictions, while a lower $\tau$ allows accurate predictions with fewer learned features. Experiments in Appendix B Figure 11 and 12 support this interpretation.

**Remark 2.1.** We believe our analysis can be extended to scenarios where feature strength varies across data by treating the set of features as a multiset, where multiple instances of the same element are allowed. The analyses in these cases are nearly identical to ours; therefore, we assume all features have identical strengths for notational simplicity.

## 3 Warm-Starting versus Cold-Starting, and a New Ideal Method

### 3.1 Experiments with Expanding Dataset

In this section, we set up the scenario where the dataset grows after each experiment in our learning framework, allowing us to compare warm-start, cold-start, and a new ideal method, which will be defined later in Sections 3.3 and 3.4.

To better understand the loss of plasticity under stationary data distribution, we consider an extreme form of stationarity where the frequency of each feature combination remains constant in each

chunk of additional data. We investigate if the loss of plasticity can manifest even under this strong stationarity. The detailed description of the dataset across the entire experiment is as follows:

**Assumption 3.1.** In each $j$-th experiment, we are provided with a training dataset $\mathcal{T}_j :=$ $\{(\boldsymbol{x}_{i,j}, y_{i,j})\}_{i \in [n]}$ with $n$ samples. For each class $c \in [C]$ and each possible feature combination $\mathcal{A} \subset \mathcal{S}_c$, we assume that $\mathcal{T}_j$ contains exactly $n_{\mathcal{A}} \geq 1$ data points with associated feature set $\mathcal{A}$, where the values of $n_{\mathcal{A}}$ are independent of $j$. Note that $\sum_{c \in [C], \mathcal{A} \subset \mathcal{S}_c} n_{\mathcal{A}} = n$. In the $j$-th experiment, we use the cumulative dataset $\mathcal{T}_{1:j} := \bigcup_{l \in [j]} \mathcal{T}_l$, the set of all training data up to the $j$-th experiment.

**Remark 3.2.** In each experiment, the feature combinations remain the same across the dataset, but the individual data points differ. This is because each data point is associated with its specific noise, which varies across samples. Although the underlying features are the same, the noise component of each data point is unique. This approach ensures that the model is exposed to a diverse set of samples.

We define a technical term $h(v; \mathcal{L}) \triangleq \frac{1}{n} \sum_{c \in [C], \mathcal{A} \subset \mathcal{S}_c} n_{\mathcal{A}} \cdot \mathbb{1} \left( v \in \mathcal{A} \wedge |\mathcal{A} \cap \mathcal{L}| < \tau \right)$ to denote the portion of data points containing $v$ not-well-classified by feature set $\mathcal{L}$. This leads to assumption:

**Assumption 3.3.** For any learned feature set $\mathcal{L} \subset \mathcal{S}$, if $v_1, v_2 \in \mathcal{S}_c$ for some class $c \in [C]$ and $h(v_1; \mathcal{L}) = h(v_2; \mathcal{L})$, then $v_1 = v_2$. Also, for any class $c \in [C]$, there exists some $\tau$ distinct features $v_1, \ldots, v_\tau \in \mathcal{S}_c$ such that $g(v_1; \mathcal{T}_j, \mathcal{T}_j), \ldots, g(v_{\tau-1}; \mathcal{T}_j, \mathcal{T}_j) \geq \gamma/n$ and $g(v_\tau; \mathcal{T}_j, \mathcal{T}_j) < \gamma/n$.

This assumption leads to Lemma D.2, stating that the order in which features are learned within a class is deterministic. This is just for simplicity of presentation and can be relaxed. The last assumption is justified by the moderate number of data points in each chunk $\mathcal{T}_j$, ensuring the existence of both $\tau - 1$ learnable features and a non-learnable feature within a class. Throughout the following discussion, we will proceed under the above assumptions unless otherwise specified.

**Notation.** We denote a model at step $s$ of the $j$-th experiment as $f^{(j,s)}$. We denote the set of learned features and the set of memorized data for the model $f^{(j,s)}$ as $\mathcal{L}^{(j,s)}$ and $\mathcal{M}^{(j,s)}$, respectively. We also define the set of data points with non-zero gradients at step $s$ of the $j$-th experiment as $\mathcal{N}^{(j,s)}$. We define respective versions of these sets and the model, with different initialization methods, denoted by the subscripts (e.g., $f_{\text{warm}}^{(j,s)}$, $f_{\text{cold}}^{(j,s)}$, and $f_{\text{ideal}}^{(j,s)}$). We emphasize that each method initializes $f^{(j,0)}, \mathcal{L}^{(j,0)}, \mathcal{M}^{(j,0)}$, and $\mathcal{N}^{(j,0)}$ differently at the start of the $j$-th experiment.

### 3.2 Prediction Process and Training Time

We provide a comparison of three initialization methods based on test accuracy and training time. To evaluate these metrics within our framework, we define the prediction process and training time.

**Prediction Process.** The model predicts unseen data points by comparing the learned features with features present in a given data point $\boldsymbol{x}$. If the overlap between the learned feature set $\mathcal{L}$ and the features in $\boldsymbol{x}$, denoted as $\mathcal{V}(\boldsymbol{x})$, is at least $\tau$, i.e., $|\mathcal{V}(\boldsymbol{x}) \cap \mathcal{L}| \geq \tau$, the model correctly classifies the data point. Otherwise, the model resorts to random guessing.

**Training Time.** Accurately measuring training time within our discrete learning framework is challenging. To address this, we introduce an alternative for training time of $j$-th experiment: the number of training data points with non-zero gradients at the start of $j$-th experiment, $|\mathcal{N}^{(j,0)}|$. This represents the amount of "learning" required for the model to classify all data points correctly. We empirically validated this proxy in practical scenarios, as shown in Figures 6 and 7 in Appendix B.1. Additionally, Nakkiran et al. (2021) observe that in real-world neural network training, when other components are fixed, the training time increases with the number of data points to learn.

### 3.3 Comparison Between Warm-Starting and Cold-Starting in Our Framework

Now we analyze the warm-start and cold-start initialization methods within our framework, focusing on test accuracy and training time. We note that, by definition, $\mathcal{L}_{\text{cold}}^{(j,0)}$ and $\mathcal{M}_{\text{cold}}^{(j,0)}$ are both empty sets, while $\mathcal{L}_{\text{warm}}^{(j,0)} = \mathcal{L}_{\text{warm}}^{(j-1,s_{j-1})}$ and $\mathcal{M}_{\text{warm}}^{(j,0)} = \mathcal{M}_{\text{warm}}^{(j-1,s_{j-1})}$, where $s_j$ denotes the last step of $j$-th experiment. Besides, we use a shorthand notation for step $s_j$ of the experiment $j$ that we drop $s$ if $s = s_j$ (e.g., $\mathcal{L}^{(j)} := \mathcal{L}^{(j,s_j)}$). For the detailed algorithms based on our learning framework, see Algorithms 3 and 4 in Appendix E.

In the test data, a feature combination $\mathcal{A} \subset \mathcal{S}_c$ of data point with class $c \in [C]$ appears with probability $n_{\mathcal{A}}/n$ along with data-specific noise. By Section 3.2, test accuracy for a learned set $\mathcal{L}$ and training time are defined as:

$$\mathrm{ACC}(\mathcal{L}) \triangleq 1 - \frac{C-1}{C} \cdot \frac{1}{n} \sum_{c \in [C], \mathcal{A} \subset \mathcal{S}_c} n_{\mathcal{A}} \cdot \mathbb{1}\left(|\mathcal{A} \cap \mathcal{L}| < \tau\right)$$

$$T_{\mathrm{warm}}^{(J)} \triangleq \sum_{j \in [J]} \left|\mathcal{N}_{\mathrm{warm}}^{(j,0)}\right|, \ T_{\mathrm{cold}}^{(J)} \triangleq \sum_{j \in [J]} \left|\mathcal{N}_{\mathrm{cold}}^{(j,0)}\right|$$

Based on these definitions, the following theorem holds:

**Theorem 3.4.** *There exists nonempty $\mathcal{G} \subsetneq \mathcal{S}$ such that we always obtain $\mathcal{L}_{\mathrm{warm}}^{(1)} = \mathcal{L}_{\mathrm{cold}}^{(1)} = \mathcal{G}$. For all $J \geq 2$, the following inequalities hold:*

$$\mathrm{ACC}\left(\mathcal{L}_{\mathrm{warm}}^{(J)}\right) \leq \mathrm{ACC}\left(\mathcal{L}_{\mathrm{cold}}^{(J)}\right), \quad T_{\mathrm{warm}}^{(J)} < T_{\mathrm{cold}}^{(J)}$$

*Furthermore, $\mathrm{ACC}(\mathcal{L}_{\mathrm{warm}}^{(J)}) < \mathrm{ACC}(\mathcal{L}_{\mathrm{cold}}^{(J)})$ holds when $J > \frac{\gamma}{\delta n}$ where $\delta \triangleq \max_{v \in \mathcal{S} \setminus \mathcal{G}} h(v; \mathcal{G}) > 0$.*

*Proof Idea.* After the first experiment, the data points in $\mathcal{T}_1$ cannot further contribute to the learning process of the warm-started model. Consequently, even when a new data chunk is provided in subsequent experiments, the feature frequencies are too small, resulting in a weak signal strength of features that cannot overcome the noise signal strength. As a result, the model memorizes individual noise components of the new data points. This procedure is repeated with every experiment, causing the learned feature set to remain *the same* as at the end of the first experiment. In contrast, when receiving $\mathcal{T}_{1:j}$ at once (cold-starting), the signal strength of features is large enough to overcome the noise signal strength, allowing the model to learn many more features. $\qquad\square$

Theorem 3.4 highlights a trade-off between cold-starting and warm-starting. Regarding test accuracy, the theorem concludes that cold-starting can achieve strictly higher accuracy than warm-starting. However, warm-starting requires a strictly shorter training time compared to cold-starting.

Detailed proof is provided in Appendix D. Theorem 3.4 suggests that the loss of plasticity in the incremental setting under the stationary assumption can be attributed to the noise memorization process. A similar observation is made in real-world neural network training. It is widely believed that during the early stages of training, neural networks primarily focus on learning features from the dataset, and after learning these features, the model starts to memorize data points that it fails to classify correctly using the learned features. To investigate this phenomenon, we conducted an experiment where CIFAR-10 was divided into two chunks, each containing 50% of the training dataset. The model was pre-trained on one chunk and then further trained on the full dataset for 300 epochs. We used three-layer MLP and ResNet-18 with SGD optimizer across 10 random seeds.

Figure 2 shows the change in the model's performance based on the duration of pre-training. When pre-training is stopped at a certain epoch and the model is then trained on the full dataset, test accuracy is maintained. However, if pre-training continues beyond a specific threshold (approximately 50% pre-training accuracy in this case), warm-starting significantly impairs the model's performance as it increasingly memorizes training data points. We attribute this phenomenon to the neural network's memorization process after learning features. This is consistent with reports of a critical learning period where neural networks learn useful features in the early phase of learning (Achille et al., 2018; Frankle et al., 2020; Kleinman et al., 2024), and with findings that neural networks tend to learn features followed by memorizing noises (Arpit et al., 2017; Jiang et al., 2020). Using the same experimental settings as in Figure 2, we tested with a large-scale dataset, ImageNet-1k, and observed similar trends (see Figure 8 in Appendix B).

**Remark 3.5.** Igl et al. (2020) find that training a model on random labels followed by corrected labels results in better generalization compared to pre-training on a subset of correctly labeled data and then further training on the full dataset with the same distribution. Achille et al. (2018) also observe that pre-training with slightly blurred images followed by original images yields worse test accuracy than pre-training with random label or random noise images. These findings align with our observations: re-training with corrected labels after random label learning "revives" gradients for most memorized data points, enabling new feature learning. Conversely, with static distributions, gradients for memorized data points remain suppressed, leading to learning from only a few data points with active gradients, causing memorization.

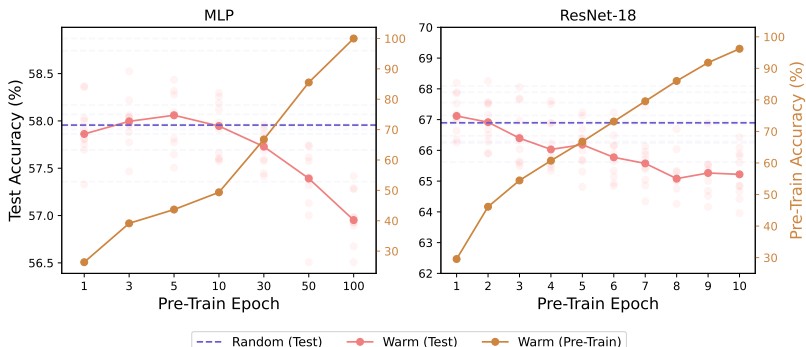

Figure 2: The plot shows the test accuracy (left y-axis) when the model is pretrained for varying epochs (x-axis) and then fine-tuned on the full data, along with the pretrain accuracy (right y-axis) plotted in brown. We trained three-layer MLP (left) and ResNet-18 (right). Each transparent line and point corresponds to a specific random seed, and the median values are highlighted with opaque markers and solid lines. The 'Random' corresponds to training from random initialization (cold-start).

## 3.4 An Ideal Method: Retaining Features and Forgetting Noise

In Section 3.3, we observed a trade-off between warm-starting and cold-starting. Cold-starting often achieves better test accuracy compared to warm-starting, while warm-starting requires less time to converge. The results suggest that neither retaining all learned information nor discarding all learned information is ideal. To address this trade-off and get the best of both worlds, we consider an ideal algorithm where we retain all learned features while forgetting all memorized data points. For any experiment $J \geq 2$, if we consider the ideal initialization, learned features $\mathcal{L}_{\text{ideal}}^{(J-1)}$ are retained, and memorized data points $\mathcal{M}_{\text{ideal}}^{(J-1)}$ are reset to an empty set. Pseudo-code for this method is given in Algorithm 5, which can be found in Appendix E. We define $T_{\text{ideal}}^{(J)} \triangleq \sum_{j \in [J]} \left| \mathcal{N}_{\text{ideal}}^{(j,0)} \right|$ as the training time with the ideal method, where $\mathcal{N}_{\text{ideal}}^{(j,0)}$ represents the set of data points having a non-zero gradient at the initial step of the $j$-th experiment. Then, we have the following theorem:

**Theorem 3.6.** *For any experiment $J \geq 2$, the following holds:*

$$\text{ACC}\left(\mathcal{L}_{\text{cold}}^{(J)}\right) = \text{ACC}\left(\mathcal{L}_{\text{ideal}}^{(J)}\right), \quad T_{\text{warm}}^{(J)} < T_{\text{ideal}}^{(J)} < T_{\text{cold}}^{(J)}$$

The detailed proof is provided in Appendix D. The ideal algorithm addresses the trade-off between cold-starting and warm-starting. We conducted an experiment to investigate the performance gap between these initialization methods.

**Synthetic Experiment.** To verify our theoretical findings in more realistic scenarios, we conducted an experiment that more closely resembles real-world settings. Instead of fixing the frequency of each feature set, we sampled each feature's existence from a Bernoulli distribution to construct $\mathcal{V}(\boldsymbol{x})$. This ensures that the experiment is more representative of real-world scenarios. Specifically, for each data point $(\boldsymbol{x}, y)$, we uniformly sampled $y \in \{0, 1\}$. From the feature set $\mathcal{S}_y$ corresponding to the sampled class $y$, we sampled features where each feature's existence follows a Bernoulli distribution, $\mathbb{1}\left(v_{y,k} \in \mathcal{V}(\boldsymbol{x})\right) \sim \text{Ber}(p_k)$, for all $v_{y,k} \in \mathcal{S}_y$. This approach allows us to model the variability in feature occurrence that is commonly observed in real-world datasets while still maintaining the core principles of our learning framework. We set the number of features, $K = 50$, with $p_k$ sampled from a uniform distribution, $\text{U}(0, 0.2)$. Each chunk contained 1000 data points with total 50 experiments, with $\gamma = 50, \tau = 3$. We sampled 10000 test data from the same distribution.

As shown in Figure 3, the results align with the above theorems. Random initialization, i.e. cold-starting, and ideal initialization achieve almost identical generalization performance, outperforming warm initialization. However, with warm initialization, the model converges faster, as evidenced by the number of non-zero gradient data points, which serves as a proxy for training time. Ideal initialization requires less time compared to cold-starting, which is also consistent with Theorem 3.6. Due to the sampling process in our experiment, we observe a gradual increase in the number of learned features and test accuracy in warm-starting, mirroring real-world observations. These findings remained robust across diverse hyperparameter settings (see Figures 9–11 in the Appendix B).

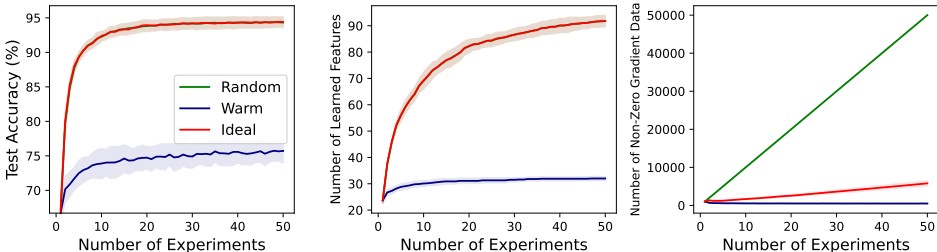

Figure 3: Comparison of random, warm, and ideal methods across 10 random seeds (mean ± std dev). The test accuracy (left) and the number of learned features across all classes (middle) are nearly identical for random and ideal initializations, causing their plots to overlap. Warm initialization, however, exhibits lower test accuracy compared to both methods. Regarding training time (right), there is a significant gap between random and warm initialization, which the ideal method addresses.

# 4 DASH: Direction-Aware SHrinking

The ideal method recycles memorized training samples by forgetting noise while retaining learned features. From now on, we shift our focus to a practical scenario: training neural networks with real-world data. This brings up the question of whether such an ideal approach can be applied in real-world settings. To address this, we propose our algorithm, **Direction-Aware SHrinking (DASH)**, which intuitively captures this idea in practical training scenarios. The outlined behavior is illustrated in Figure 4. When new data is introduced, DASH shrinks each weight based on its alignment with the negative gradient of the loss calculated from the training data, placing more emphasis on recent data.

If the degree of alignment is small (i.e., the cosine similarity is close to or below 0), we consider that the weight has not learned a proper feature and shrinks it significantly to make it "forget" learned information. This allows weights to forget memorized noises and easily change their direction. On the other hand, if the weight and negative gradient are well-aligned (i.e., the cosine similarity is close to 1), we consider it learned features and we shrink the weight to a lesser degree to maintain the learned information. This method aligns with the intuition of the ideal method, as it allows us to shrink weights that have not learned proper information while retaining weights that have learned commonly observed features.

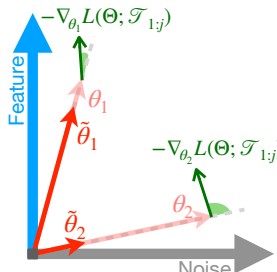

Figure 4: Illustration of DASH. We compute the loss $L$ with training data $\mathcal{T}_{1:j}$ and obtain the negative gradient. Then, we shrink the weights proportionally to the cosine similarity between the current weight $\theta$ and $\nabla_\theta L$, resulting in $\tilde{\theta}$.

---

**Algorithm 1** Direction-Aware SHrinking (DASH)

**Require:**
- Model $f_\Theta$ with list of parameters $\Theta$ after the $(j-1)$-th experiment
- Training data points $\mathcal{T}_{1:j}$
- Averaging coefficient $0 < \alpha \le 1$
- Threshold $\lambda > 0$

1: **Initialize:**
   $\quad G_\theta^{(0)} \leftarrow 0, \forall \theta \text{ in } \Theta$
2: **for** $i$ in $1:j$ **do**
3: $\quad \ell \leftarrow \text{Loss}(f_\Theta, \mathcal{T}_i)$
4: $\quad U_\Theta \leftarrow \text{Gradient of loss } \ell$
5: $\quad$ **for** $\theta$ in $\Theta$ **do**
6: $\quad\quad G_\theta^{(i)} \leftarrow (1-\alpha) \times G_\theta^{(i-1)} + \alpha \times U_\theta$
7: $\quad$ **end for**
8: **end for**
9: **for** $\theta$ in $\Theta$ **do**
10: $\quad s_\theta \leftarrow \text{CosineSimilarity}\left(-G_\theta^{(j)}, \theta\right)$
11: $\quad \theta \leftarrow \theta \odot \max\{\lambda, s_\theta\}$
12: **end for**
13: **return** model $f_\Theta$, initialized for the $j$-th experiment

---

The shrinking is done *per neuron*, where the incoming weights are grouped into a weight vector denoted as $\theta$. For convolutional filters, the height and width of the kernel are flattened to form a single weight vector $\theta$ for each pair of input and output filters. DASH has two hyperparameters: $\lambda$ and $\alpha$. Hyperparameter $\lambda$ is the minimum shrinkage threshold, as each weight vector is shrunk by $\max\{\lambda, \text{cos\_sim}\}$, while $\alpha$ denotes the coefficient of exponential moving average of per-chunk loss gradients. Lower $\alpha$ value gives more weight to previous gradients, resulting in less shrinkage. This

is because gradients of the previously seen data usually have high cosine similarity with learned weights; low $\alpha$ is advantageous for simpler datasets where preserving learned features helps. Note that DASH is an initialization method applied *only once* when new data is introduced. The detailed algorithm is presented in Algorithm 1.

We discuss our intuition regarding the connection between DASH and the spirit of the ideal method. Generally, features from previous data are likely to reappear frequently in new data, as they are relevant to the data class. In contrast, noise from previous data rarely reappears as it is not class-specific and varies with each data point. As a result, the negative gradient of the loss naturally aligns more closely with learning features rather than memorizing noise from older data. This suggests that when neurons are aligned with its negative gradient of the loss, we can assume they have learned important features and should not be shrunk.

To validate our intuition, we plotted the accuracy on previously learned data points a few epochs after applying DASH in Figure 13, Appendix B. Our experiments show that DASH recovers training accuracy on previous datasets more quickly than other methods, likely because it preserves learned features while discarding memorized noise. As experiments progress, the growing number of learned features allows DASH to retain more information, leading to improved training accuracy across successive experiments. We further validate our intuition behind DASH in Figure 14 in Appendix B.

## 5   Experiments

### 5.1   Experimental Details

Our setup is similar to the one described in Ash and Adams (2020). We divided the training dataset into 50 chunks, and at the beginning of each experiment, a chunk is added to the existing training data. Models were considered converged and each experiment was terminated when training accuracy reached 99.9%, aligning with our learning framework. We conducted experiments with vanilla training i.e. without data augmentations, weight decay, learning rate schedule, etc. Appendix C.4 presents additional results on other settings, including the state-of-the-art (SoTA) settings that include the techniques mentioned above. We evaluated DASH on Tiny-ImageNet, CIFAR-10, CIFAR-100, and SVHN using ResNet-18, VGG-16, and three-layer MLP architectures with batch normalization layer. Models were trained using Stochastic Gradient Descent (SGD) and Sharpness-Aware Minimization (SAM) (Foret et al., 2020), both with momentum.

DASH was compared against baselines (cold-starting, warm-starting, and S&P (Ash and Adams, 2020)) and methods addressing plasticity loss under non-stationarity (L2 INIT (Kumar et al., 2023) and Reset (Nakkiran et al., 2021)). Layer normalization (Ba et al., 2016) and SAM (Foret et al., 2020), known to mitigate plasticity loss in reinforcement learning (Lee et al., 2023), were applied to both warm and cold-starting. Consistent hyperparameters were used across all methods, with details provided in Appendix C.3. S&P, Reset, and DASH were applied whenever new data was introduced. We report two metrics for both test accuracy and number of steps required for convergence: the value from the final experiment and the average across all experiments.

### 5.2   Experimental Results

We first experimented with CIFAR-10 on ResNet-18 to determine if methods from previous works for mitigating plasticity based on non-stationarity can be a solution to our incremental setting with stationarity. Appendix C.1 shows that L2 INIT, Reset, layer normalization, and reviving dead neurons, are not effective in our setting. Thus, we conducted the remaining experiments without these methods. Additionally, Table 1 shows that warm-starting with SAM does not outperform cold-starting with SAM, indicating that SAM alone is not an effective method in our case. Table 1 shows that DASH surpasses cold-starting (Random Init) and S&P in most cases. Training times were often shorter compared to training from scratch, and when longer, the performance gap in test accuracy was more pronounced. Omitted results are in Tables 3-6 located in Appendix C.2. Additionally, we confirm that DASH is computationally efficient, with details on the computation and memory overhead comparisons provided in Appendix C.5.

We argue that S&P can cause the model to forget learned information, including important features, due to shrinking every weight uniformly and perturbing weights. This leads to increased training time and relatively lower test accuracy, especially in SoTA settings (see Appendix C.4.1). In contrast, DASH addresses these issues by preserving learned features with direction-aware weight shrinkage.

Table 1: Results of training with various datasets using ResNet-18. Bold values indicate the best performance. For the number of steps, bold formatting is used for all methods *except* warm-starting. Results are averaged across five random seeds, except for Tiny-ImageNet which uses three random seeds. Standard deviations are provided in parentheses.

| ResNet-18 | Test Acc at Last Experiment | | Number of Steps at Last Experiment | | AVG of Test Acc across All Experiments | | AVG of Number of Steps across All Experiments | |
|---|---|---|---|---|---|---|---|---|
| *T-ImageNet* | SGD | SAM | SGD | SAM | SGD | SAM | SGD | SAM |
| Random Init | 25.69 (0.13) | 31.30 (0.09) | 30237 (368) | 40142 (368) | 17.37 (0.06) | 21.95 (0.11) | 17503 (53) | 22513 (74) |
| Warm Init | 9.57 (0.24) | 13.94 (0.37) | 3388 (368) | 5474 (0) | 6.70 (0.04) | 9.88 (0.21) | 1785 (5) | 2773 (7) |
| S&P | 34.34 (0.48) | 37.39 (0.18) | 13815 (368) | 26066 (1606) | 25.43 (0.02) | 28.47 (0.08) | 7940 (15) | 13172 (182) |
| DASH | **46.11 (0.34)** | **49.57 (0.36)** | **8341 (368)** | **12251 (368)** | **33.06 (0.15)** | **35.93 (0.17)** | **4439 (48)** | **7900 (136)** |
| *CIFAR-10* | | | | | | | | |
| Random Init | 67.32 (0.51) | 75.68 (0.39) | **5161 (156)** | 17125 (292) | 57.66 (0.11) | 66.27 (0.13) | 2916 (37) | **8121 (26)** |
| Warm Init | 63.53 (0.56) | 70.99 (0.59) | 1173 (0) | 3910 (247) | 54.87 (0.18) | 63.27 (0.55) | 665 (11) | 2153 (23) |
| S&P | 81.25 (0.14) | 85.53 (0.22) | 5395 (625) | 32649 (978) | 71.74 (0.16) | 76.19 (0.04) | 2766 (53) | 15552 (1558) |
| DASH | **84.08 (0.52)** | **86.75 (0.53)** | 6490 (399) | 11886 (2771) | **75.21 (0.33)** | **77.59 (0.69)** | 3454 (55) | 8689 (527) |
| *CIFAR-100* | | | | | | | | |
| Random Init | 35.52 (0.14) | 40.27 (0.31) | 10557 (247) | 14310 (191) | 25.72 (0.11) | 29.90 (0.06) | 5803 (79) | 7588 (54) |
| Warm Init | 25.12 (0.59) | 32.02 (0.31) | 1173 (0) | 2346 (0) | 19.18 (0.52) | 24.01 (0.33) | 854 (23) | 1294 (12) |
| S&P | 50.08 (0.23) | 52.95 (0.36) | 4926 (191) | 12277 (1226) | 37.32 (0.14) | 40.36 (0.18) | 2929 (27) | **5954 (187)** |
| DASH | **57.99 (0.28)** | **60.88 (0.29)** | **3519 (0)** | 11730 (1211) | **43.99 (0.14)** | **46.15 (0.58)** | 2041 (51) | 6675 (797) |
| *SVHN* | | | | | | | | |
| Random Init | 86.27 (0.46) | 89.84 (0.24) | 5552 (156) | **10869 (156)** | 78.01 (0.10) | 83.31 (0.14) | 3099 (15) | **5546 (44)** |
| Warm Init | 84.01 (0.41) | 88.85 (0.29) | 938 (191) | 1329 (191) | 75.37 (0.50) | 81.16 (0.54) | 642 (18) | 993 (15) |
| S&P | 92.67 (0.17) | 94.27 (0.07) | **3597 (156)** | 1573 (191) | 87.35 (0.14) | 89.35 (0.05) | **1858 (12)** | 5548 (94) |
| DASH | **93.67 (0.13)** | **95.19 (0.09)** | 5161 (672) | 14467 (989) | **89.59 (0.07)** | **91.67 (0.03)** | 2619 (68) | 8613 (728) |

Theorem 3.6 shows that ideal initialization can achieve the same test accuracy as cold-starting. Yet in practice, DASH surpasses cold-starting in test accuracy. This could be due to the difference between the discrete learning process in our framework and the continuous learning process in real-world neural network training. Even if features have already been learned, DASH can learn them in greater strength compared to learning from scratch by preserving previously learned features during training.

Further insights into the applicability of DASH can be found in Appendix C.4. We evaluate DASH in various settings beyond our original setup. In the SoTA setting, different observations are made: DASH achieves test accuracy close to (but does not outperform) cold-starting, without requiring additional hyperparameter tuning, which aligns more closely with our theoretical analysis. We demonstrate DASH's scalability on large-scale datasets such as ImageNet-1k. We also examine two additional practical scenarios: a data-discarding setting and a situation where new data are continuously added. In such cases, applying DASH with an interval (rather than upon every arrival of new data) proves effective. Finally, we explore DASH's behavior in non-stationary environments, specifically in Class Incremental Learning (CIL) with data accumulation settings.

## 6   Discussion and Conclusion

In this work, we defined an abstract framework for feature learning and discovered that warm-starting benefits from reduced training time compared to random initialization but can hurt the generalization performance of neural networks due to the memorization of noise. Motivated by these observations, we proposed Direction-Aware SHrinking (DASH), which shrinks weights that learned data-specific noise while retaining weights that learned commonly appearing features. We validated DASH in real-world model training, achieving promising results for both test accuracy and training time.

Loss of plasticity is problematic in situations where new data is continuously added daily, which is the case in many real-world application scenarios. Our research aimed to interpret and resolve this issue, preventing substantial waste of energy, time, and the environment. By elucidating the loss of plasticity phenomenon in stationary data distributions, we have taken a crucial step towards addressing challenges that may emerge in real-world AI, where the continuous influx of additional data is inevitable.

We hope our fundamental analysis of the loss of plasticity phenomenon sheds light on understanding this issue as well as providing a remedy. To generalize our findings to any neural network architecture, we treated the learning process as a discrete abstract procedure and did not assume any hypothesis class. Future research could focus on understanding the loss of plasticity phenomenon via optimization or theoretically analyzing it in non-stationary data distributions, such as in reinforcement learning.

## Acknowledgement

This work was partly supported by a National Research Foundation of Korea (NRF) grant (No. RS-2024-00421203) funded by the Korean government (MSIT), and an Institute for Information & communications Technology Planning & Evaluation (IITP) grant (No. RS-2022-II220184, Development and Study of AI Technologies to Inexpensively Conform to Evolving Policy on Ethics) funded by the Korean government (MSIT). This research was supported in part by the NAVER-Intel Co-Lab. The work was conducted by KAIST and reviewed by both NAVER and Intel.

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

# Contents

# A    Detailed Intuition and Example of Our Framework

In this section, we provide a detailed explanation of how our theoretical framework reflects the intuitive process of learning from image data. Our framework is designed to capture the characteristics of image data, where the input includes both relevant information for the image labels (which we refer to as "*features*" such as a cow's ears, eyes, tail, and mouth in Figure 5a) and irrelevant information (which we refer to as "*noise*" such as the sky or grass circled in red in Figure 5a). Our framework builds on the insights from Shen et al. (2022) and incorporates them into a discrete learning model.

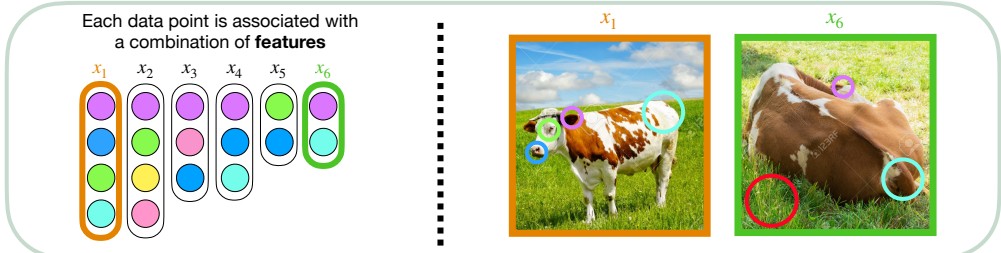

(a) Examples of data points in the feature learning framework. Data points $x_1$ through $x_6$ denotes each data point composed of a combination of features (left side of the dashed line). For instance, the first data point on the left ($x_1$) can correspond to the features of the first cow image on the right, while the last data point on the left ($x_6$) can correspond to the features of the second cow image on the right. All data points also contain data-specific noise, such as the sky or grass in the cow images, circled in red in the cow image on the right.

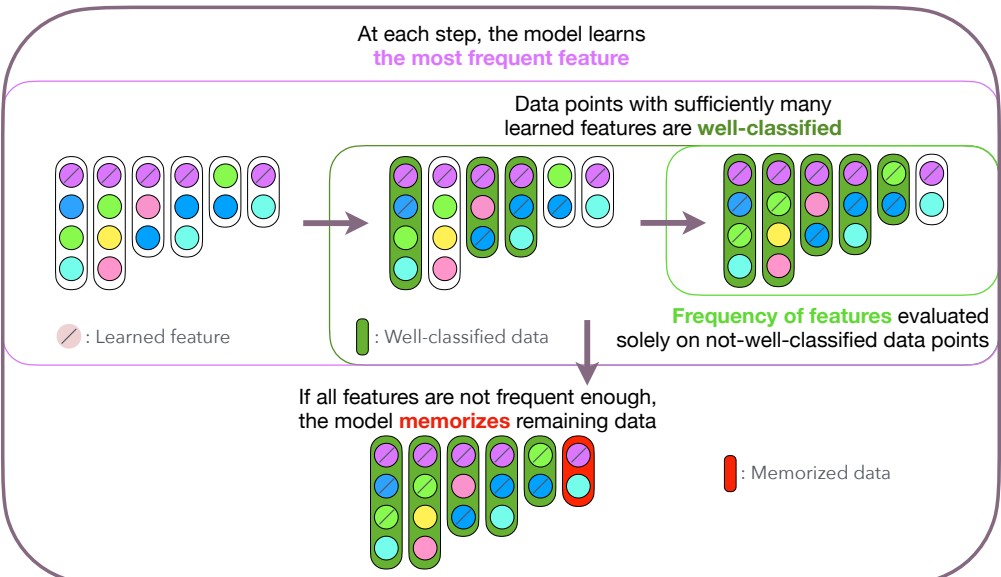

(b) An illustration of our feature learning process framework. The model sequentially learns the most frequent features from the not-well-classified data points, and if the features are insufficient to be learned properly, the process ends by memorizing the noise. With $\tau = 2$ and $\gamma = 2$, the model learns three features in sequence (violet, blue, green). The remaining data point which corresponds to $x_6$ in Figure 5a is memorized with noise.

Figure 5: An illustration of our proposed feature learning framework with single class. Each data point is composed of features, as shown in Figure 5a, and is learned through the framework depicted in Figure 5b.

Before explaining the intuition of our learning framework, let us connect notation from Section 2 with Figure 5. Figure 5 shows six data points from a single cow class, where training set $\mathcal{T} = \{(\boldsymbol{x}_i, y)\}_{i \in [6]}$. Features are denoted by their color's first letter: violet as v, blue as b, green as g, mint as m, yellow as y, and pink as p. The complete feature set $\mathcal{S}$ is therefore $\{v, b, g, m, y, p\}$. The learned feature set $\mathcal{L}$ is empty at initialization, and the set of data points with non-zero gradients $\mathcal{N} \subset \mathcal{T}$ is equal to $\mathcal{T}$ at initialization. We set the data learning threshold $\tau = 2$ and noise strength $\gamma = 2$.

Our training process is based on the idea that features which occur more frequently in the training data are easier to learn, as the gradients tend to align more strongly with these frequent features. Consequently, our framework is designed to learn the most frequent features in a sequential manner. For example, a cow's ears, shown in violet in Figure 5 would be selected first since it is the most frequent feature:

$$\mathrm{v} = \arg\max_{c \in \mathcal{S} \setminus \mathcal{L}^{(0)}} g(c; \mathcal{T}, \mathcal{N}^{(0)}) = \arg\max_{c \in \{v,b,g,m,y,p\}} g(c; \mathcal{T}, \mathcal{T})$$

where $\mathcal{L}^{(s)}$ and $\mathcal{N}^{(s)}$ represent the learned feature set and the set of data points with non-zero gradients, respectively, at step $s$. This holds because:

$$g(\mathrm{v}; \mathcal{T}, \mathcal{N}^{(0)}) = \frac{1}{|\mathcal{T}|} \sum_{(\boldsymbol{x},y) \in \mathcal{N}^{(0)}} \mathbb{1}(\mathrm{v} \in \mathcal{V}(x)) = \frac{5}{6}$$
$$> g(c; \mathcal{T}, \mathcal{N}^{(0)})$$

for all $c \in \{b, g, m, y, p\}$. Since we have set $\gamma$ to 2, the feature v is learned because its frequency is sufficiently high compared to the noise, i.e., $g(\mathrm{v}; \mathcal{T}, \mathcal{N}^{(0)}) \geq \gamma/|\mathcal{T}| = \frac{2}{6}$. Also, since $\tau$ set to 2, no data point is considered well-classified at this stage, i.e., $|\{v\} \cap \mathcal{V}(\boldsymbol{x}_i)| < \tau$ for all $i \in [6]$. As a result, $\mathcal{L}^{(1)} = \{v\}, \mathcal{N}^{(1)} = \mathcal{T}$.

In the next step, the model learns feature b in the same way. Therefore, $\boldsymbol{x}_1$, $\boldsymbol{x}_3$, and $\boldsymbol{x}_4$ are now considered well-classified since they share two overlapping features with the learned feature set $\mathcal{L}^{(2)} = \{v, b\}$. This indicates that the model accumulates enough information to accurately classify these cow images as cows. Once the model correctly classifies data based on these features, the influence of those data points on the learning process diminishes, as the loss gradients become smaller as the predictions become more confident. To account for this, our framework evaluates the frequency of features only on data points that are not yet well-classified. Therefore, the set of data points with non-zero gradients is now updated to $\mathcal{N}^{(2)} = \{(\boldsymbol{x}_i, y)\}_{i \in \{2,5,6\}}$. Subsequently, we compute the gradient proxy $g(c; \mathcal{T}, \mathcal{N}^{(2)})$ using only the data points in $\mathcal{N}^{(2)}$. In the third step, considering the features included in $\{\boldsymbol{x}_2, \boldsymbol{x}_5, \boldsymbol{x}_6\}$, we get that the feature g is the most frequent among those in $\mathcal{S} \setminus \mathcal{L}^{(2)}$, and the feature meets the threshold for learning. Hence, the feature g is newly learned, leading us to $\mathcal{L}^{(3)} = \{v, b, g\}$ and $\mathcal{N}^{(3)} = \{(\boldsymbol{x}_6, y)\}$.

As training progresses, the algorithm may reach a point where the remaining features in the not-well-classified data points are too infrequent to be learned effectively. At this point, the noise in each data point has a faster learning speed, which leads the model to memorize noise instead of learning the features to achieve 100% training accuracy. In the example described in Figure 5, at the fourth step, the remaining feature m in $\boldsymbol{x}_6$ does not meet the threshold: $g(\mathrm{m}; \mathcal{T}, \mathcal{N}^{(3)}) = 1/6 < \gamma/|\mathcal{T}|$, so the model is unable to learn feature m. Consequently, the model shifts to memorizing the noise in the remaining data point $\boldsymbol{x}_6$ and completes training by memorizing the noise.

## B  Connection Between Our Framework and Practical Scenarios

This section demonstrates how our theoretical framework relates to real-world scenarios. Section B.1 presents real-world experimental evidence supporting our theoretical framework. Section B.2 explores the robustness of our synthetic experiments across various hyperparameters. Finally, Section B.3 provides empirical validation of the key intuitions behind DASH.

### B.1  Justification of Our Framework

Figure 6 shows that the initial gradient norm of training data, $|\mathcal{N}^{(j,0)}|$, can be a proxy for the training time until convergence. As the initial gradient norm increases, the number of steps required for convergence also increases. While this figure uses the gradient norm instead of the number of non-zero gradient data points due to the continuous nature of real-world neural network training, we believe it resembles the behavior of non-zero gradient data points. Additionally, Figure 7 demonstrates that the number of steps required for convergence increases as the number of data points increases in various datasets.

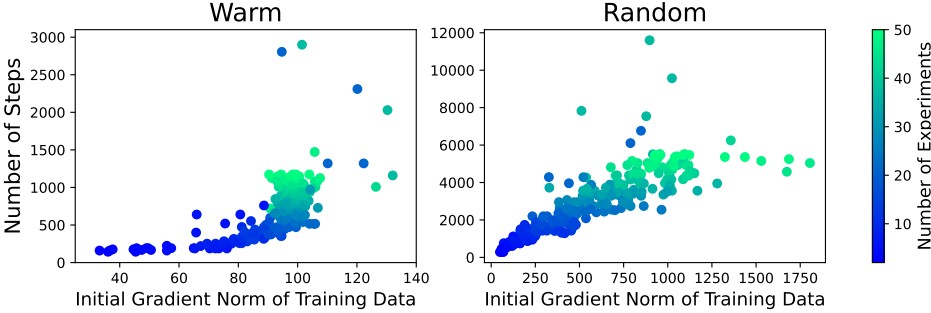

Figure 6: Trained on ResNet-18 with five random seeds, where CIFAR-10 is divided into 50 chunks and incrementally increased by adding new chunks at each experiment. Each point represents an individual experiment. The gradient norm is used as a proxy for the number of non-zero gradient data points, which in turn serves as a proxy for the training time. A larger gradient norm indicates the model needs to learn more features or memorize more data points to correctly classify all training data points.

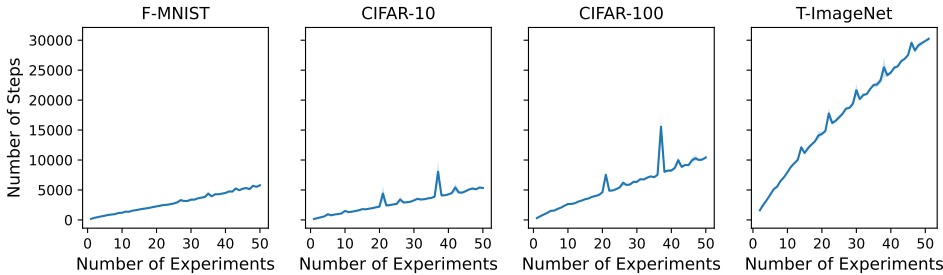

Figure 7: Figure trained on ResNet-18 with three random seeds. The dataset is divided into 50 chunks, and new chunks are incrementally added for each experiment. The number of steps required for convergence increases with the amount of data when training a cold-started neural network, which is a standard training.

We also investigate the effect of the pretrain epoch on warm-starting in ImageNet-1k classification with ResNet18, similar to the setup in Figure 2 in Section 3.3. We conducted experiments using different pretrain epochs: 30, 50, 70, and 150. Figure shows a declining trend in accuracy for warm-started models as pretrain epoch increases. Interestingly, we noted a similar phenomenon in Figure 2 in Section 3.3, where warm-starting does not negatively impact test accuracy when training resumes from earlier pretrained epochs, such as 30 in this case. This observation aligns with our

theoretical framework, which suggests that neural networks tend to learn meaningful features first and then begin to memorize noise later. We believe this phenomenon is persistent across different datasets, model architectures, and optimizers.

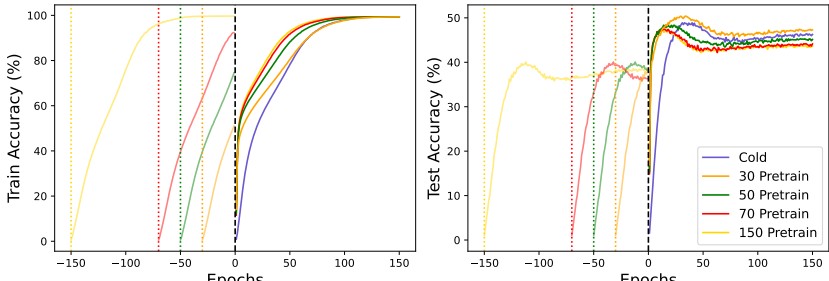

Figure 8: Effect of pretraining epochs on warm-starting with the large-scale ImageNet-1k dataset using ResNet-18. The plots to the left of the dashed line represent the pretraining results, and the dotted lines indicate the starting points for each pretraining run. Test accuracy declines as the number of pretraining epochs increases, particularly beyond 50 epochs, which aligns with the observations in Figure 2 in Section 3.3.

## B.2 Further Results on Warm-starting vs. Cold-starting

We conducted synthetic experiments across a wide range of hyperparameters. Figure '9 uses the same setup as Section 3.4 but varies numbers of classes, $C$. Figure 10 investigates varying noise signal strengths, $\gamma$, while Figure 11 explores different values of $\tau$. These results align with our findings from Theorems 3.4 and 3.6.

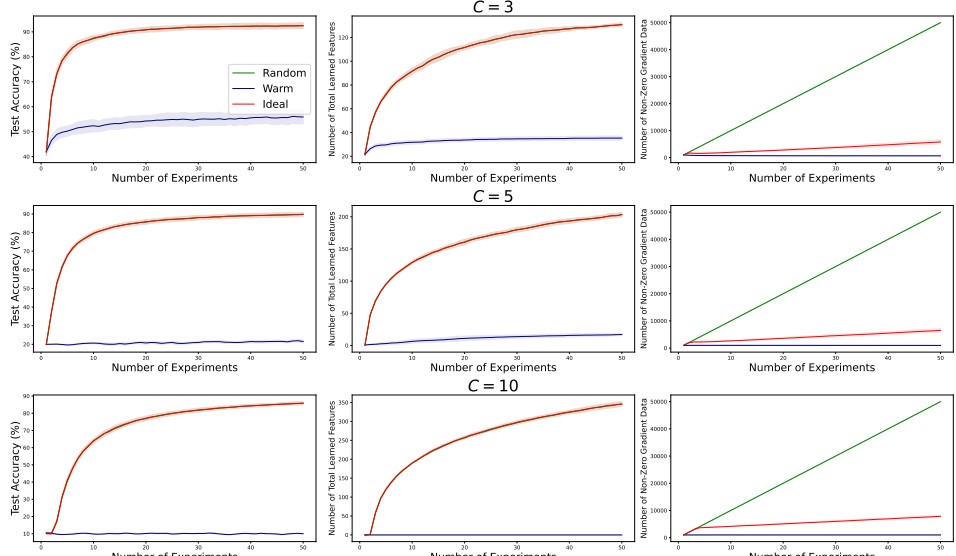

Figure 9: Results using the same hyperparameters as described in Section 3.4, except for the number of classes ($C$). Experiments were conducted with 10 random seeds. The trend observed in Figure 3 persists across different values of $C$.

As stated in Section 2, we posited that $\tau$ could serve as a proxy for dataset complexity. Figure 11 shows that as $\tau$ increases, the threshold for considering a data point well-classified also increases, making it more difficult to correctly predict unseen data points. This difficulty is particularly pronounced for warm-starting, leading to a widening gap between the random initialization and warm initialization methods. Additionally, this phenomenon is observed in real-world neural network training, as depicted in Figure 12. For datasets with higher complexity (from left to right), the gap between the two initialization methods widens, exhibiting the same trend as an increasing $\tau$.

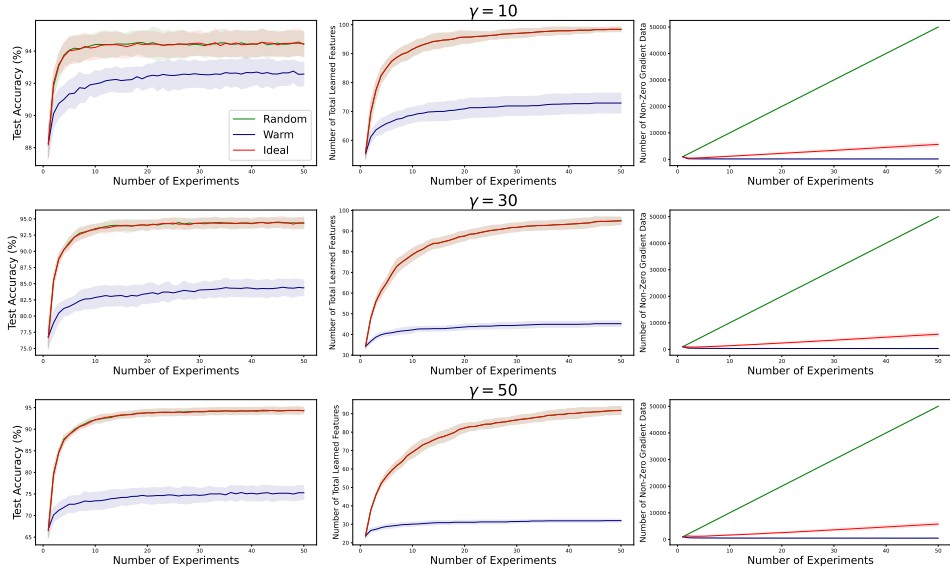

Figure 10: Results using the same hyperparameters as described in Section 3, except for the strength of the noise, $\gamma$. Experiments were conducted with 10 random seeds. The trend observed in Figure 3 persists across different values of $\gamma$.

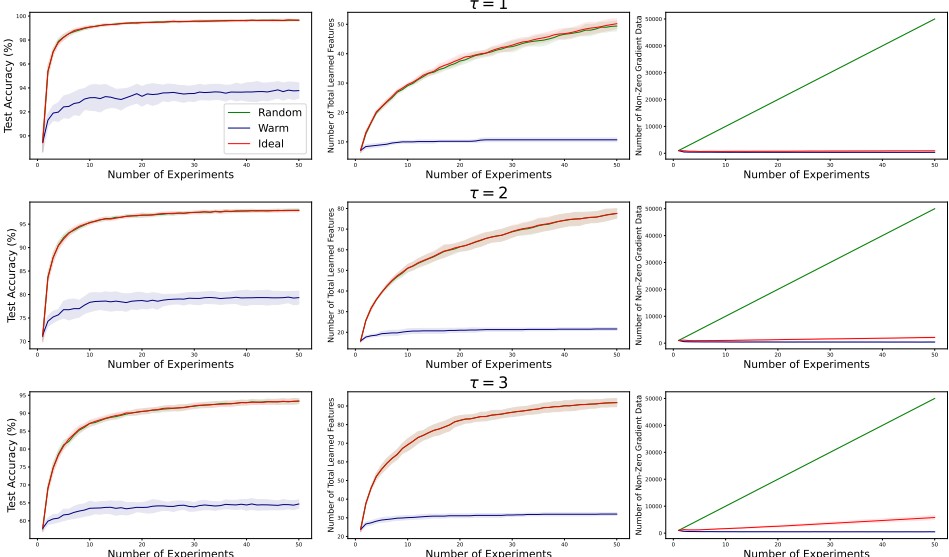

Figure 11: Results using the same hyperparameters as described in Section 3, except for the threshold for a data point is considered well-classified ($\tau$). Experiments were conducted with 10 random seeds. The trend observed in Figure 3 persists across different values of $\tau$.

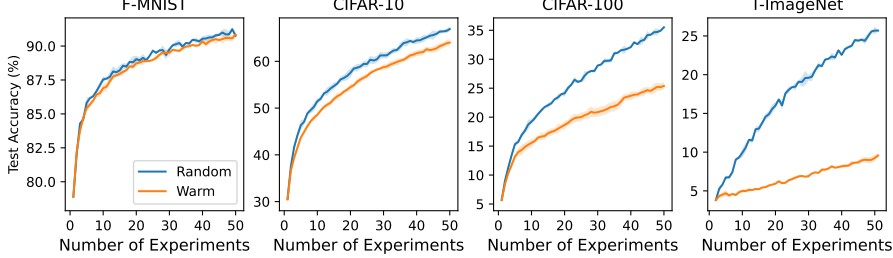

Figure 12: The same hyperparameters are used described in Section 5.1 with three random seeds. The gap in test accuracy between the two initialization methods increases as dataset complexity increases.

## B.3   Experiments Supporting the Intuition behind DASH

To validate whether previously learned/memorized data points do not have large gradients when further trained with a combined dataset (existing + newly introduced data) using warm-starting, we plotted the train accuracy on the previous dataset for the first few epochs. Using ResNet-18 on CIFAR-10 (Figure 13), we found that warm-starting preserves performance on previously learned data even when training continues with combined datasets, supporting the main idea of Theorem 3.4.

To verify whether DASH truly captures our intuitions from the ideal algorithms, we conducted an experiment using CIFAR-10 trained on ResNet-18, with the same experimental settings. Figure 13 demonstrates that when applying DASH, the train accuracy on previous datasets increases more rapidly after a few epochs compared to other methods. We argue that this behavior stems from our algorithm's ability to forget memorized noise while preserving learned features. As the number of experiments increases, the number of learned features also grows. For a fair comparison, we used $\lambda = 0.05$ for DASH, and when performing S&P and shrink, we shrank each weight by multiplying 0.05. In the case of S&P, after shrinking, we added noise sampled from $\mathcal{N}(0, 0.01^2)$.

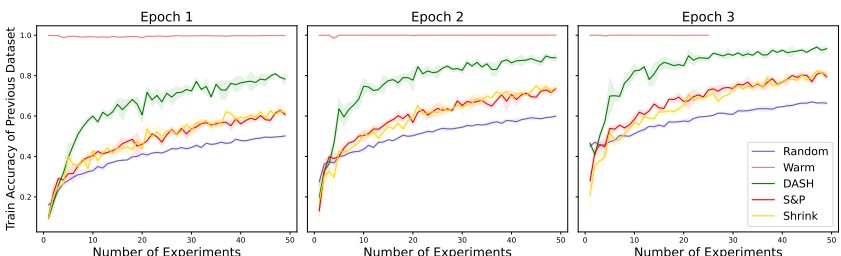

Figure 13: The results are averaged over 10 random seeds. The x-axis represents the number of experiments, while the y-axis represents the training accuracy on previous datasets. Warm-starting can retain previously learned data points when further trained with an incremented dataset. Additionally, DASH, plotted in green, can retain more information compared to other methods.

We further validate our intuition that cosine similarity between the negative loss gradient and model weights can indicate whether the model has indeed learned features. We trained a 3-layer CNN on CIFAR-10, varying the size of the training dataset. We observed that the cosine similarity between the negative gradient from the test data and the learned filters increases as the training dataset size grows. The model trained with more data appears to learn more features, as evidenced by the rising trend in test accuracy, shown by the dashed line in Figure 14. This suggests that cosine similarity can capture whether the weights have learned features, which aligns with our expectations.

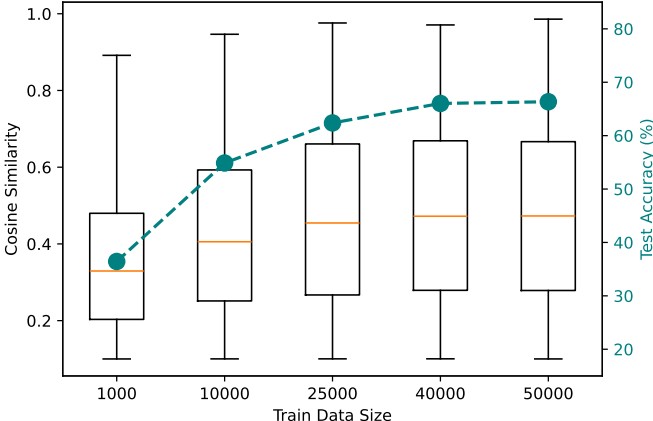

Figure 14: We trained a 3-layer CNN on CIFAR-10 and plotted cosine similarity values greater than 0.1. The left y-axis (black) represents the cosine similarity between the negative loss gradient from the training data and the model filters, while the right y-axis (green) shows the test accuracy.

# C    Omitted Experimental Results

This section presents additional experimental results and their corresponding hyperparameters. Appendix C.1 examines non-stationary solutions, demonstrating their limitations in stationary settings. The following section compares DASH against other baselines in various settings, excluding those methods from Appendix C.1 due to their poor performance in stationary conditions. Appendix C.3 provides a comprehensive list of hyperparameters used across all experiments from Section 5, as well as Appendix C.1 and C.2. Additionally, we validate DASH's applicability including state-of-the-art (SoTA) settings in Appendix C.4, and analyze computational and memory overhead in Appendix C.5.

We ran our experiments on two distinct hardware setups. The first server had an Intel Gaudi-v2 HPU with 96GB of VRAM with four Intel Xeon Platinum 8380 40-core CPUs. The second server had an NVIDIA A6000 GPU with 48GB of VRAM, paired with two AMD EPYC 7763 64-core CPUs. We compared the computing performance of these two hardware setups in Appendix C.6.

Unless specified otherwise, we follow the experimental settings outlined in Section 5.1. We conducted an experiment with an incremental training dataset comprised of 50 chunks. At the start of each experiment, a new chunk is provided and added to the existing training dataset. Before proceeding to the next experiment, the model is trained until achieving 99.9% train accuracy.

## C.1    Methods for Non-Stationary Data Distribution Struggle in Stationary Settings

In this subsection, we describe solutions that aim to mitigate plasticity loss under non-stationarity, which cannot remedy the loss of plasticity in an incremental setting with a stationary data distribution. Table 2 shows L2 INIT (Kumar et al., 2023) and Reset (Nikishin et al., 2022) cannot be a solution in our setting.

Table 2: Results of training CIFAR-10 dataset trained on various models with solutions proposed to mitigating loss of plasticity in non-stationary data distributions. Bold values indicate the best performance. For the number of steps, we did not provide bold formatting. Results are averaged over three random seeds, with standard deviations provided in parentheses.

| CIFAR-10 | Test Acc at last experiment | | Number of Steps at last experiment | | AVG of Test Acc across all experiments | | AVG of Number of Steps across all experiments | |
|---|---|---|---|---|---|---|---|---|
| *ResNet-18* | SGD | SAM | SGD | SAM | SGD | SAM | SGD | SAM |
| Random Init | **66.75 (0.55)** | **75.55 (0.18)** | 5213 (184) | 17734 (184) | **57.82 (0.04)** | **66.19 (0.01)** | 2889 (24) | 8100 (7) |
| Warm Init | 64.10 (0.12) | 70.56 (0.30) | 1173 (0) | 4040 (184) | 55.11 (0.10) | 62.94 (0.47) | 726 (29) | 2160 (11) |
| L2 INIT | 64.24 (0.80) | 70.32 (0.09) | 1173 (0) | 4040 (184) | 55.47 (0.43) | 62.55 (0.19) | 648 (14) | 2139 (15) |
| Reset | 63.97 (0.45) | 72.03 (0.33) | 1173 (0) | 17986 (1596) | 55.55 (0.30) | 63.40 (0.26) | 976 (51) | 7225 (10) |
| | | | | | | | | |
| *VGG-16* | | | | | | | | |
| Random Init | **84.19 (0.35)** | **86.64 (0.12)** | 21375 (1475) | 37032 (1243) | **75.62 (0.08)** | **77.01 (0.22)** | 12743 (280) | 12509 (343) |
| Warm Init | 78.93 (0.44) | 82.04 (0.04) | 1825 (184) | 4692 (319) | 70.62 (0.24) | 74.00 (0.33) | 1954 (42) | 4277 (315) |
| L2 INIT | 82.79 (0.04) | 82.11 (0.19) | 193936 (58167) | 6126 (665) | 72.11 (0.14) | 73.77 (0.37) | 12489 (443) | 4390 (94) |
| Reset | 78.71 (0.26) | 81.88 (0.35) | 1564 (0) | 3910 (552) | 70.45 (0.36) | 73.31 (0.25) | 1814 (30) | 3230 (51) |
| | | | | | | | | |
| *MLP* | | | | | | | | |
| Random Init | **57.54 (0.31)** | **58.62 (0.13)** | 13555 (184) | 19289 (184) | **51.23 (0.42)** | 52.02 (0.24) | 7516 (166) | 9794 (127) |
| Warm Init | 56.44 (0.33) | 57.67 (0.45) | 2346 (0) | 2216 (184) | 50.60 (0.41) | 51.98 (0.14) | 2309 (408) | 1701 (34) |
| L2 INIT | 56.38 (0.39) | 58.24 (0.06) | 1955 (0) | 2085 (184) | 50.56 (0.51) | **52.15 (0.33)** | 2221 (423) | 1604 (73) |
| Reset | 53.82 (0.32) | 56.42 (0.09) | 6125 (487) | 3389 (184) | 48.89 (0.24) | 50.70 (0.25) | 5955 (740) | 2465 (64) |

Furthermore, applying layer normalization cannot close the gap between cold-starting and warm-starting; rather, the gap increases, as shown in Figure 15. Also, Nikishin et al. (2022) and Sokar et al. (2023) state that loss of plasticity in non-stationary data distributions arises from inactive neurons in the model. However, this is not the case in our setting, as demonstrated in Figure 16.

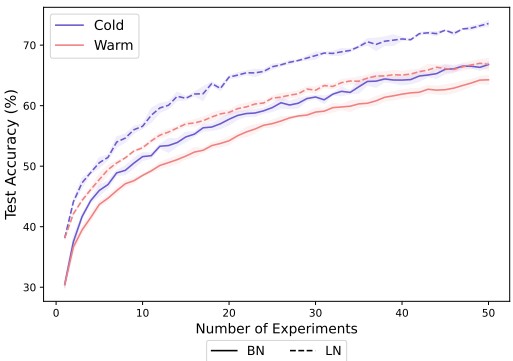

Figure 15: The figure shows the results of training ResNet-18 on CIFAR-10 with three random seeds. Layer normalization (dashed lines) is applied in place of batch normalization in ResNet-18, while solid lines represent the use of standard batch normalization. The red lines denote warm-starting, and the blue lines denote cold-starting. The figure demonstrates that the layer normalization technique cannot serve as a solution for plasticity loss. Moreover, the gap between warm-starting and cold-starting performance increases when layer normalization is employed.

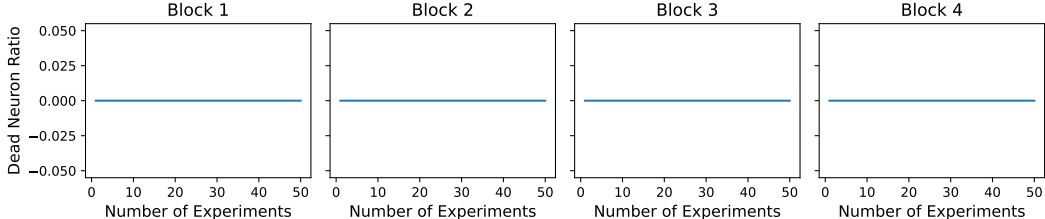

Figure 16: The figure presents the results of training ResNet-18 on CIFAR-10 with three random seeds. The presence of dead neurons is assessed after each block of ResNet-18 with training dataset, and the analysis reveals that there are no dead neurons. This finding suggests that techniques designed to revive dead neurons in non-stationary data distributions cannot effectively address the plasticity loss observed in the incremental learning setting with stationary data, which is the primary focus of our study.

## C.2 Results of Experiments Across Different Datasets, Models, and Optimizers

This subsection presents our complete experimental results (Tables 3-6), following the settings detailed in Section 5.1. We evaluated multiple approaches, including "Warm ReM" (warm-starting with momentum reset), which also proved ineffective. We conducted comprehensive comparisons across multiple datasets (Tiny-ImageNet, CIFAR-10, CIFAR-100, and SVHN), architectures (ResNet-18, VGG-16, three-layer MLP), and optimizers (SGD and SGD-based SAM). The results demonstrate that DASH consistently outperforms baseline methods - warm-starting, cold-starting, and S&P - while often requiring less training time. Results are averaged across five random seeds, except for Tiny-ImageNet which uses three random seeds due to its high computational cost.

Table 3: Average results from three training runs using various models on the Tiny-ImageNet dataset.

| T-ImageNet | Test Acc at last experiment | | Number of Steps at last experiment | | AVG of Test Acc across all experiments | | AVG of Number of Steps across all experiments | |
|---|---|---|---|---|---|---|---|---|
| ResNet-18 | SGD | SAM | SGD | SAM | SGD | SAM | SGD | SAM |
| Random Init | 25.69 (0.13) | 31.30 (0.09) | 30237 (368) | 40142 (368) | 17.37 (0.06) | 21.95 (0.11) | 17503 (53) | 22513 (74) |
| Warm Init | 9.57 (0.24) | 13.94 (0.37) | 3388 (368) | 5474 (0) | 6.70 (0.04) | 9.88 (0.21) | 1785 (5) | 2773 (7) |
| Warm ReM | 9.20 (0.16) | 13.71 (0.29) | 3388 (368) | 5474 (0) | 6.67 (0.08) | 9.93 (0.30) | 1787 (17) | 2795 (14) |
| S&P | 34.34 (0.48) | 37.39 (0.18) | 13815 (368) | 26066 (1606) | 25.43 (0.02) | 28.47 (0.08) | 7940 (15) | 13172 (182) |
| DASH | **46.11 (0.34)** | **49.57 (0.36)** | **8341 (368)** | **12251 (368)** | **33.06 (0.15)** | **35.93 (0.17)** | **4439 (48)** | **7900 (136)** |
| VGG-16 | | | | | | | | |
| Random Init | 40.26 (0.30) | 42.41 (0.13) | 92927 (8940) | 29976 (664) | 28.19 (0.03) | **30.40 (0.04)** | 48878 (799) | 17094 (192) |
| Warm Init | 17.11 (0.44) | 20.77 (0.32) | 1955 (0) | 2997 (184) | 12.91 (0.18) | 15.14 (0.35) | 4359 (162) | 2513 (13) |
| Warm ReM | 17.51 (0.38) | 20.23 (0.06) | 2085 (184) | 2867 (184) | 12.97 (0.24) | 14.87 (0.14) | 4130 (99) | 2472 (8) |
| S&P | 36.56 (0.96) | 38.63 (0.73) | **59432 (5538)** | **18898 (368)** | 23.91 (0.09) | 25.98 (0.22) | **28747 (366)** | **10494 (45)** |
| DASH | **44.29 (0.55)** | **44.40 (0.19)** | 69989 (6215) | 22938 (1329) | **28.47 (0.49)** | 29.11 (0.73) | 31864 (362) | 14258 (149) |
| MLP | | | | | | | | |
| Random Init | 9.12 (0.06) | 9.19 (0.25) | 28934 (0) | 42749 (975) | 6.94 (0.01) | 7.22 (0.03) | **13596 (35)** | **17871 (71)** |
| Warm Init | 7.44 (0.18) | 7.74 (0.25) | 4692 (0) | 4952 (368) | 6.18 (0.03) | 6.41 (0.11) | 2437 (17) | 2797 (38) |
| Warm ReM | 7.54 (0.19) | 7.86 (0.08) | 4431 (368) | 5474 (0) | 6.34 (0.04) | 6.23 (0.05) | 2411 (35) | 2821 (36) |
| S&P | 9.61 (0.22) | 10.28 (0.25) | 33365 (2879) | 55782 (975) | 7.27 (0.01) | 7.57 (0.04) | 16227 (1458) | 21126 (94) |
| DASH | **10.17 (0.19)** | **10.77 (0.12)** | 30237 (975) | 47702 (638) | **7.67 (0.02)** | **8.12 (0.03)** | 17743 (899) | 19455 (212) |

Table 4: Average results from five training runs using various models on the CIFAR-10 dataset.

| CIFAR-10 | Test Acc at last experiment | | Number of Steps at last experiment | | AVG of Test Acc across all experiments | | AVG of Number of Steps across all experiments | |
|---|---|---|---|---|---|---|---|---|
| ResNet-18 | SGD | SAM | SGD | SAM | SGD | SAM | SGD | SAM |
| Random Init | 67.32 (0.51) | 75.68 (0.39) | **5161 (156)** | 17125 (292) | 57.66 (0.11) | 66.27 (0.13) | 2916 (37) | **8121 (26)** |
| Warm Init | 63.53 (0.56) | 70.99 (0.59) | 1173 (0) | 3910 (247) | 54.87 (0.18) | 63.27 (0.55) | 665 (11) | 2153 (23) |
| Warm ReM | 63.96 (0.64) | 70.82 (0.36) | 1173 (0) | 3988 (292) | 55.03 (0.47) | 63.17 (0.60) | 703 (49) | 2158 (4) |
| S&P | 81.25 (0.14) | 85.53 (0.22) | 5395 (625) | 32649 (978) | 71.74 (0.16) | 76.19 (0.04) | **2766 (53)** | 15552 (1558) |
| DASH | **84.08 (0.52)** | **86.75 (0.53)** | 6490 (399) | **11886 (2771)** | **75.21 (0.33)** | **77.59 (0.69)** | 3454 (55) | 8689 (527) |
| VGG-16 | | | | | | | | |
| Random Init | 84.11 (0.32) | 84.67 (0.12) | 23225 (2565) | 21270 (14166) | 75.64 (0.16) | 75.77 (1.81) | 12723 (233) | **9358 (4306)** |
| Warm Init | 79.01 (0.45) | 82.09 (0.16) | 2111 (530) | 4770 (455) | 70.90 (0.41) | 74.03 (0.26) | 1950 (34) | 4180 (271) |
| Warm ReM | 78.82 (0.32) | 81.66 (0.44) | 2737 (2358) | 4532 (312) | 71.23 (0.31) | 73.43 (0.59) | 2056 (111) | 4051 (63) |
| S&P | 84.96 (0.46) | 88.02 (0.21) | 21426 (672) | **34251 (10994)** | 76.62 (0.16) | 79.13 (0.19) | 11812 (180) | 14452 (751) |
| DASH | **87.57 (0.26)** | **90.68 (0.26)** | **17008 (2150)** | 45668 (15184) | **79.63 (0.32)** | **83.07 (0.20)** | **10171 (266)** | 20814 (6416) |
| MLP | | | | | | | | |
| Random Init | **57.42 (0.31)** | 58.53 (0.55) | 13528 (191) | 19315 (191) | 51.09 (0.37) | 51.94 (0.23) | **7598 (167)** | 9770 (113) |
| Warm Init | 56.28 (0.42) | 57.60 (0.37) | 2346 (0) | 2189 (191) | 50.39 (0.41) | 51.82 (0.23) | 2195 (352) | 1706 (27) |
| Warm ReM | 56.07 (0.37) | 57.71 (0.42) | 2189 (191) | 2111 (191) | 50.25 (0.44) | 51.81 (0.26) | 2519 (351) | 1650 (54) |
| S&P | 57.02 (0.40) | 58.19 (0.52) | 6647 (428) | 7038 (349) | 50.87 (0.26) | 51.93 (0.18) | 7939 (969) | 4188 (112) |
| DASH | 57.41 (0.48) | **58.60 (0.36)** | **6021 (681)** | **6021 (398)** | **51.20 (0.25)** | **52.26 (0.43)** | 7821 (1308) | **3772 (66)** |

Table 5: Average results from five training runs using various models on the CIFAR-100 dataset.

| CIFAR-100 | Test Acc at last experiment | | Number of Steps at last experiment | | AVG of Test Acc across all experiments | | AVG of Number of Steps across all experiments | |
|---|---|---|---|---|---|---|---|---|
| ResNet-18 | SGD | SAM | SGD | SAM | SGD | SAM | SGD | SAM |
| Random Init | 35.52 (0.14) | 40.27 (0.31) | 10557 (247) | 14310 (191) | 25.72 (0.11) | 29.90 (0.06) | 5803 (79) | 7588 (54) |
| Warm Init | 25.12 (0.59) | 32.02 (0.31) | 1173 (0) | 2346 (0) | 19.18 (0.52) | 24.01 (0.33) | 854 (23) | 1294 (12) |
| Warm ReM | 24.83 (0.61) | 31.63 (0.58) | 1173 (0) | 2346 (0) | 18.98 (0.57) | 23.75 (0.41) | 822 (27) | 1291 (14) |
| S&P | 50.08 (0.23) | 52.95 (0.36) | 4926 (191) | 12277 (1226) | 37.32 (0.14) | 40.36 (0.18) | 2929 (27) | **5954 (187)** |
| DASH | **57.99 (0.28)** | **60.88 (0.29)** | **3519 (0)** | **11730 (1211)** | **43.99 (0.14)** | **46.15 (0.58)** | **2041 (51)** | 6675 (797) |
| VGG-16 | | | | | | | | |
| Random Init | 54.03 (0.45) | 57.29 (2.29) | 62560 (5251) | 26900 (1512) | 39.78 (0.11) | 42.39 (1.70) | 29436 (477) | 18107 (1295) |
| Warm Init | 37.14 (1.22) | 39.91 (0.58) | 3362 (191) | 4379 (292) | 28.98 (0.99) | 30.07 (0.51) | 4196 (216) | 3482 (227) |
| Warm ReM | 38.21 (0.81) | 39.58 (0.72) | 3362 (191) | 3988 (156) | 28.98 (0.99) | 30.58 (0.49) | 4196 (216) | 3251 (58) |
| S&P | 59.61 (0.43) | **63.67 (0.44)** | **29637 (834)** | **11573 (635)** | **45.36 (0.17)** | **47.92 (0.18)** | 14329 (149) | **7644 (197)** |
| DASH | **59.79 (0.28)** | 61.91 (1.10) | 53109 (11451) | 26275 (5716) | 44.01 (0.34) | 45.38 (0.68) | 26577 (4650) | 16163 (4461) |
| MLP | | | | | | | | |
| Random Init | 28.25 (0.40) | 29.46 (0.34) | 17516 (292) | 25571 (725) | 22.39 (0.11) | 23.53 (0.10) | **13245 (2270)** | **12467 (301)** |
| Warm Init | 26.20 (0.34) | 27.45 (0.14) | 3449 (156) | 3128 (247) | 21.56 (0.12) | 22.47 (0.07) | 5461 (1089) | 2793 (158) |
| Warm ReM | 26.14 (0.26) | 27.54 (0.46) | 4144 (635) | 3362 (191) | 21.41 (0.05) | 22.57 (0.11) | 8422 (3059) | 2866 (144) |
| S&P | 30.12 (0.27) | 30.04 (0.20) | **10948 (428)** | 23851 (1504) | **23.44 (0.13)** | 23.79 (0.09) | 37317 (6536) | 14873 (810) |
| DASH | **30.13 (0.35)** | **31.22 (0.44)** | 16578 (635) | **22052 (805)** | 23.42 (0.08) | **24.43 (0.08)** | 52328 (3918) | 13408 (579) |

Table 6: Average results from five training runs using various models on the SVHN dataset.

| SVHN | Test Acc at last experiment | | Number of Steps at last experiment | | AVG of Test Acc across all experiments | | AVG of Number of Steps across all experiments | |
|---|---|---|---|---|---|---|---|---|
| *ResNet-18* | SGD | SAM | SGD | SAM | SGD | SAM | SGD | SAM |
| Random Init | 86.27 (0.46) | 89.84 (0.24) | 5552 (156) | **10869 (156)** | 78.01 (0.10) | 83.31 (0.14) | 3099 (15) | **5546 (44)** |
| Warm Init | 84.01 (0.41) | 88.85 (0.29) | 938 (191) | 1329 (191) | 75.37 (0.50) | 81.16 (0.54) | 642 (18) | 993 (15) |
| Warm ReM | 83.85 (0.38) | 88.75 (0.27) | 782 (0) | 1485 (156) | 75.41 (0.85) | 81.03 (0.62) | 640 (6) | 1006 (13) |
| S&P | 92.67 (0.17) | 94.27 (0.07) | **3597 (156)** | 11573 (191) | 87.35 (0.14) | 89.35 (0.05) | **1858 (12)** | 5548 (94) |
| DASH | **93.67 (0.13)** | **95.19 (0.09)** | 5161 (672) | 14467 (989) | **89.59 (0.07)** | **91.67 (0.03)** | 2619 (68) | 8613 (728) |
| | | | | | | | | |
| *VGG-16* | | | | | | | | |
| Random Init | 93.65 (0.20) | 93.88 (0.17) | 16187 (1201) | 12355 (312) | 90.43 (0.09) | 90.53 (0.07) | 8617 (222) | 7379 (275) |
| Warm Init | 92.67 (0.18) | 93.08 (0.19) | 1485 (625) | 938 (191) | 89.61 (0.05) | 89.80 (0.10) | 1122 (34) | 959 (37) |
| Warm ReM | 92.85 (0.26) | 93.24 (0.17) | 1329 (191) | 1016 (191) | 89.64 (0.27) | 89.83 (0.14) | 1128 (29) | 935 (43) |
| S&P | 94.58 (0.20) | 94.83 (0.16) | **9853 (758)** | **8289 (455)** | 91.82 (0.10) | 91.94 (0.09) | **5979 (104)** | **4979 (122)** |
| DASH | **94.72 (0.19)** | **94.84 (0.20)** | 12668 (1925) | 8836 (530) | **91.84 (0.10)** | **92.05 (0.14)** | 6844 (225) | 5769 (331) |
| | | | | | | | | |
| *MLP* | | | | | | | | |
| Random Init | **82.92 (0.24)** | **83.68 (0.26)** | 31768 (1942) | 36206 (585) | **77.19 (0.13)** | **78.18 (0.06)** | 19861 (436) | 18278 (277) |
| Warm Init | 81.17 (0.17) | 82.29 (0.21) | 4789 (324) | 2893 (191) | 76.51 (0.21) | 77.55 (0.07) | 7317 (806) | 2510 (48) |
| Warm ReM | 81.21 (0.32) | 82.25 (0.04) | 4398 (507) | 3128 (319) | 76.53 (0.15) | 77.46 (0.05) | 6147 (553) | 2626 (108) |
| S&P | 82.07 (0.27) | 82.81 (0.33) | 28621 (3376) | 16734 (518) | 77.00 (0.15) | 77.94 (0.13) | **16530 (1019)** | 9802 (222)) |
| DASH | 82.30 (0.38) | 83.02 (0.26) | **25571 (1411)** | **15405 (944)** | 76.77 (0.13) | 77.89 (0.07) | 21092 (1535) | **8956 (255)** |

## C.3 Hyperparameters

In this section, we provide the details of the hyperparameters used in our experiments from Section 5, as well as Appendix C.1 and C.2. Additionally, we present heatmaps illustrating the results for a wide range of two hyperparameters, $\alpha$ and $\lambda$, in DASH. The heatmaps in Figure 17 suggest that DASH exhibits robustness to hyperparameter variations, indicating that its performance is less affected by the choice of hyperparameter values.

We fixed the momentum to 0.9 and the batch size to 128. The learning rate is set to 0.001 for training ResNet-18, and for other models, a learning rate of 0.01 is used. The value of $\rho$ for SAM is chosen based on the performance of cold-starting. The default value of $\alpha = 0.3$ is used, and we did not change this value frequently. The perturbation parameter $\sigma$ used in the Shrink & Perturb (S&P) procedure is set to 0.01, as this value is considered optimal for perturbation, as described in Ash and Adams (2020). Initially, we tested $\sigma = 0.1$ as the perturbation parameter, since Ash and Adams (2020) reported slightly better test accuracy compared to $\sigma = 0.01$ in some cases. However, we experienced significantly poorer generalization performance with $\sigma = 0.1$ compared to $\sigma = 0.01$, as shown in Figure 18. The hyperparameters used in our experiments are described in Table 7.

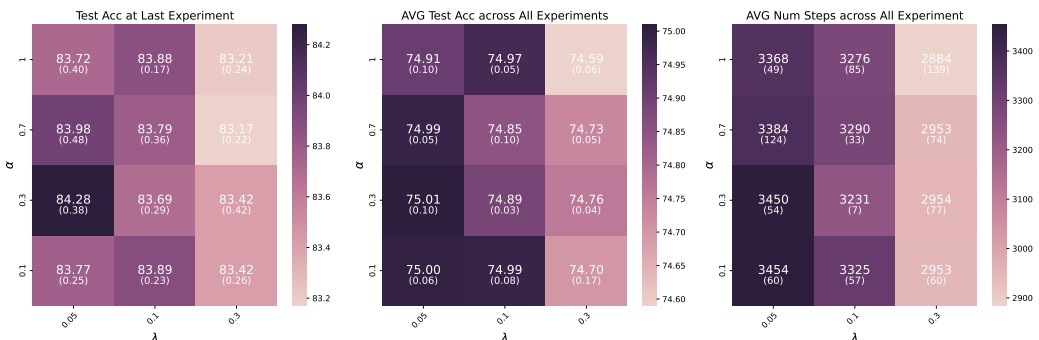

Figure 17: The performance of DASH with various hyperparameter values on the CIFAR-10 dataset using a ResNet-18 architecture. Three runs averaged with standard deviation. Darker colors indicate higher values. The first two heatmaps show that higher values are preferable, while the last heatmap demonstrates that lower values (brighter colors) are preferable.

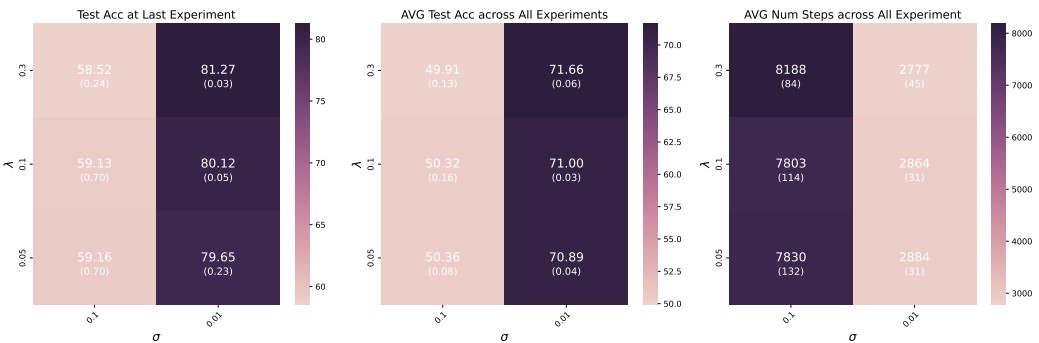

Figure 18: The performance of S&P on CIFAR-10 using ResNet-18 with varying $\sigma$ values. While Ash and Adams (2020) reported better test accuracy when $\sigma = 0.1$ compared to $\sigma = 0.01$, we exhibited significantly lower performance compared to $\sigma = 0.01$.

Table 7: The hyperparameters used in our experiments. Values before the '/' are for SGD, and values after are for SAM. In the case of S&P, the $\lambda$ value corresponds to the shrinkage parameter, while the $\sigma$ parameter controls the magnitude of the noise added to the weights. For L2 INIT, we did not perform experiments except for CIFAR-10.

| | | | | | DASH | | S&P | | L2 INIT |
|---|---|---|---|---|---|---|---|---|---|
| **ResNet-18** | Momentum | LR | Batch Size | $\rho$ | $\lambda$ | $\alpha$ | $\lambda$ | $\sigma$ | $\lambda$ |
| Tiny-Imagenet | 0.9 | 0.001 | 128 | 0.05 | 0.05 | 0.3 | 0.05 | 0.01 | - |
| CIFAR-10 | 0.9 | 0.001 | 128 | 0.1 | 0.05/0.3 | 0.3 | 0.3 | 0.01 | 1e-4 |
| CIFAR-100 | 0.9 | 0.001 | 128 | 0.05 | 0.1 | 0.3 | 0.3 | 0.01 | - |
| SVHN | 0.9 | 0.001 | 128 | 0.05 | 0.3 | 0.3 | 0.3 | 0.01 | - |
| **VGG16** | | | | | | | | | |
| Tiny-Imagenet | 0.9 | 0.01 | 128 | 0.05 | 0.05 | 0.3 | 0.05 | 0.01 | - |
| CIFAR-10 | 0.9 | 0.01 | 128 | 0.1 | 0.05/0.1 | 0.3 | 0.1 | 0.01 | 1e-4 |
| CIFAR-100 | 0.9 | 0.01 | 128 | 0.03 | 0.05 | 0.9/0.3 | 0.3 | 0.01 | - |
| SVHN | 0.9 | 0.01 | 128 | 0.01 | 0.1 | 0.9/0.3 | 0.3 | 0.01 | - |
| **MLP** | | | | | | | | | |
| Tiny-Imagenet | 0.9 | 0.01 | 128 | 0.1 | 0.1 | 1.0 | 0.3/0.1 | 0.01 | - |
| CIFAR-10 | 0.9 | 0.01 | 128 | 0.1 | 0.7/0.5 | 1.0 | 0.7/0.5 | 0.01 | 1e-4 |
| CIFAR-100 | 0.9 | 0.01 | 128 | 0.1 | 0.1 | 1.0 | 0.3/0.1 | 0.01 | - |
| SVHN | 0.9 | 0.01 | 128 | 0.1 | 0.3 | 1.0 | 0.3 | 0.01 | - |

## C.4 Discussions on the Broader Applicability of DASH

In this subsection, we explore how DASH performs in a variety of settings, including state-of-the-art (SoTA) configurations, larger datasets, and scenarios where previous data cannot be stored. We also examine how DASH compares to other methods when data is continuously introduced throughout training, as well as its effectiveness in non-stationary data distribution environments, such as Class Incremental Learning (CIL) setups.

### C.4.1 Performance in State-of-the-Art Settings

In the state-of-the-art (SoTA) setting, we employed weight decay and standard data augmentation techniques, such as horizontal flipping and random cropping. We also used a learning rate scheduler that reduces the learning rate step-wise by a factor of 0.2 at 60, 120, and 200 epochs. By applying the learning rate scheduler, there is no need to compare training time since training is completed at roughly the same epoch across all experiments. The weight decay was set to 0.0005, and the initial learning rate was set to 0.1. All other settings remain the same as mentioned above. We tested this setup on CIFAR-10 and CIFAR-100 using the ResNet-18 architecture.

The results in Table 8 show that DASH performs similarly to or slightly worse than starting from random initialization. It appears that this is partly because all hyperparameters are tuned to maximize

the performance of cold-starting, to achieve the (close-to-)SoTA test accuracy numbers. Due to the lack of computational resources, we were unable to tune hyperparameters specifically for DASH.

Furthermore, we believe this aligns more closely with our theoretical anylsis in Theorem 3.6, as the hyperparameters are tuned to allow the model to learn as many features as possible, making it difficult for DASH to outperform cold-starting.

Moreover, we observe that S&P cannot be used in these SoTA settings. We believe this is due to the nature of S&P, which shrinks all weights, while the SoTA setting is likely designed to avoid learning unuseful features, unlike the previous setting. Consequently, it is plausible that retaining learned features is more important than forgetting them, making S&P unsuitable for SoTA settings. Although DASH performs slightly worse than cold-starting, it is conceivable that it is better at retaining features compared to S&P and other warm-starting methods, resulting in better overall performance.

The gap between warm-starting and cold-starting has been significantly reduced, likely due to data augmentation techniques and the increase in learning rate when new data is introduced. Data augmentation techniques increase the amount of feature information, allowing warm-starting to learn features that vanilla training (without augmentation) cannot (Shen et al., 2022). Furthermore, as the learning rate is set to a higher value at the beginning of each new experiment, the model can forget previously memorized data points and escape spurious minima that were difficult to escape from, which is consistent with the findings of Berariu et al. (2021). Despite these improvements, a gap still exists between warm-starting and cold-starting.

Table 8: Results of training CIFAR-10, CIFAR-100 dataset trained on ResNet-18 with SoTA settings. Bold values indicate the best performance, while underlined values denote the second-best performance. For the number of steps, we did not provide bold formatting since we used learning rate scheduling. Results are averaged over three random seeds, with standard deviations provided in parentheses.

| *ResNet-18* | Test Acc at last experiment | | Number of Steps at last experiment | | AVG of Test Acc across all experiments | | AVG of Number of Steps across all experiments | |
|---|---|---|---|---|---|---|---|---|
| **CIFAR-10** | SGD | SAM | SGD | SAM | SGD | SAM | SGD | SAM |
| Random Init | **94.73 (0.14)** | **95.47 (0.17)** | 50439 (319) | 47832 (184) | **88.77 (0.04)** | 89.24 (0.15) | 24826 (62) | 23751 (34) |
| Warm Init | 94.35 (0.31) | 94.80 (0.20) | 51221 (552) | 47832 (184) | 87.94 (0.26) | 88.62 (0.57) | 23759 (57) | 21821 (174) |
| Warm ReM | 94.56 (0.25) | 95.00 (0.29) | 51612 (319) | 47962 (184) | 88.20 (0.33) | 88.56 (0.60) | 23775 (16) | 21786 (79) |
| S&P | 94.15 (0.10) | 94.73 (0.07) | 51351 (184) | 48353 (184) | 88.38 (0.03) | 89.27 (0.26) | 25369 (49) | 22805 (16) |
| DASH | 94.25 (0.25) | 95.06 (0.36) | 51872 (487) | 48223 (184) | 88.65 (0.24) | **89.34 (0.40)** | 24264 (75) | 22233 (85) |
| | | | | | | | | |
| **CIFAR-100** | | | | | | | | |
| Random Init | **75.98 (0.01)** | **76.09 (0.12)** | 63081 (184) | 56825 (184) | **61.49 (0.09)** | **61.81 (0.08)** | 27536 (194) | 25521 (91) |
| Warm Init | 74.10 (0.09) | 74.21 (0.26) | 69598 (1462) | 58128 (921) | 58.40 (0.24) | 58.44 (0.12) | 28012 (114) | 24562 (243) |
| Warm ReM | 74.05 (0.13) | 74.36 (0.13) | 68425 (1689) | 57216 (664) | 58.32 (0.24) | 58.33 (0.15) | 27965 (190) | 24534 (139) |
| S&P | 72.96 (0.34) | 73.71 (0.37) | 64775 (664) | 61387 (552) | 57.33 (0.10) | 57.68 (0.06) | 28809 (148) | 26476 (212) |
| DASH | 74.84 (0.07) | 74.98 (0.09) | 67121 (1815) | 59953 (1208) | 60.89 (0.20) | 61.29 (0.13) | 28746 (306) | 25630 (100) |

## C.4.2 Scalability on Large Datasets

We validate the scalability of DASH for larger datasets such as ImageNet-1k. However, conducting experiments for such datasets is challenging, as it would require repeating the training process 50 times until convergence—an extremely time-consuming process. As an alternative, we trained ImageNet-1K on ResNet18 using a setup similar to Figure 2 in Section 3.3. Our setup involved pretraining on 50% of the data before fine-tuning on the complete dataset. We used shrinkage parameter $\lambda = 0.3$ for both DASH and S&P. Results shown in Figure 19 demonstrate that DASH achieves superior performance compared to all baseline methods—including cold initialization, warm initialization, and S&P—both in terms of test accuracy and convergence speed. Notably, DASH achieves faster convergence and marginally better test accuracy than S&P, demonstrating its effectiveness even on challenging large-scale datasets.

## C.4.3 Effectiveness in Data-discarding Setting

To explore how our algorithm can be applied to a broader range of scenarios further, we consider situations where storing all data is not feasible (e.g., due to memory constraints), rather than accumulating and retaining data. Specifically, we divided the CIFAR-10 dataset into 50 chunks and

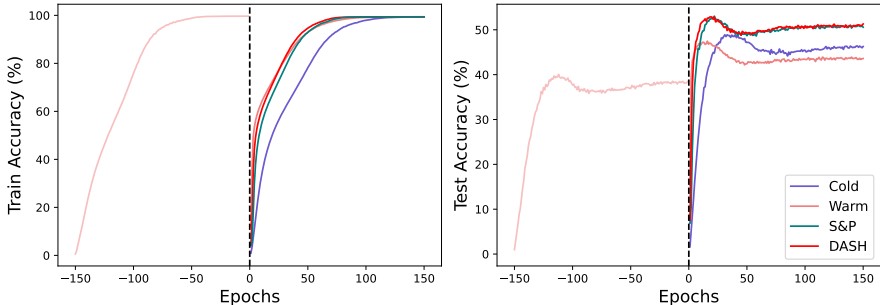

Figure 19: Evaluation of DASH on the ImageNet-1k dataset using ResNet18. We pretrained for 150 epochs using 50% of the dataset (shown to the left of the dashed line) and then continued training for another 150 epochs with the full dataset for 'Warm', 'S&P' and 'DASH' methods. For cold initialization, we trained for 150 epochs on the full dataset starting from random initialization. DASH outperforms all baseline methods in terms of test accuracy and convergence speed.

experimented with ResNet-18, running each experiment on a single chunk before moving on to the next. All other settings remained consistent with the details provided in Section 5.1.

We configured our experiments to apply DASH at specific intervals (e.g., every 10, 15, or 20 experiments) instead of after each one. Since there is no previous data available in this scenario, we set $\alpha = 0$. After the 40th experiment, when the model had sufficiently learned, we stopped applying the shrinking process. We also conducted similar experiments with the S&P method for comparison. It's important to note that this variant is feasible because both DASH and S&P focus on adjusting the model's initialization.

We tested DASH and S&P with intervals of 10, 15, and 20 epochs. For both methods, we explored shrinkage parameter ($\lambda$) of 0.05, 0.1, and 0.3. We plotted the results using the best hyperparameters for each method in Figure 20. Notably, DASH's test accuracy consistently exceeded that of S&P across all hyperparameter configurations, as shown in Figure 21. These findings demonstrate that DASH outperforms both the warm-starting baseline and the S&P method in terms of test accuracy. Based on this evidence, we can conclude that DASH is well-suited for data-discarding scenarios.

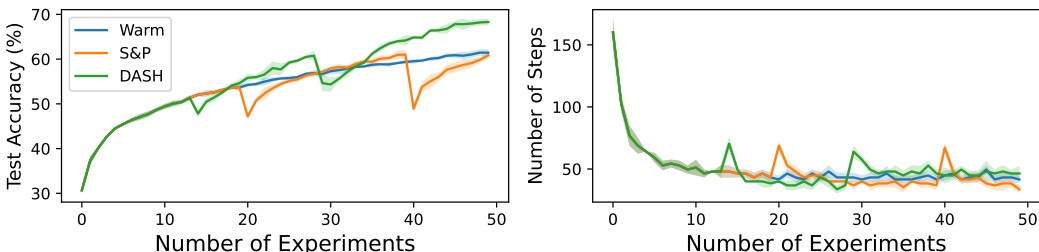

Figure 20: Performance comparison between warm-starting, S&P, and DASH in a scenario without access to previous data, using optimal hyperparameters for each method. For S&P, the shrinkage parameter $\lambda = 0.3$ with shrinkage interval 20. For DASH, the shrinkage parameter $\lambda = 0.3$ with shrinkage interval 15. DASH significantly outperforms warm-starting, while S&P performs even worse than the warm-starting baseline in terms of test accuracy. Since there are no previous data available, we observe sharp drops in test accuracy during shrinkage events, but DASH quickly recovers while S&P struggles to regain performance.

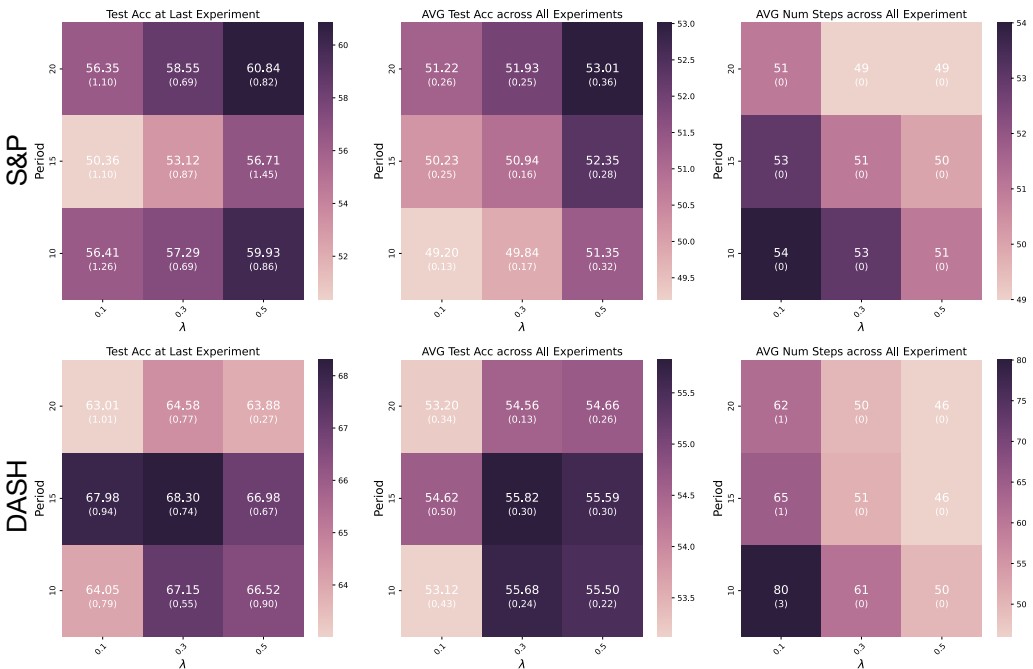

Figure 21: Heatmaps comparing S&P (top row) and DASH (bottom row) performance without access to previous data. The x-axis shows results for different shrinkage parameters $\lambda$ (0.1, 0.3, 0.5), while the y-axis shows different shrinkage periods (10, 15, 20). Left heatmaps display final test accuracy, where DASH outperforms S&P across all hyperparameter configurations. Middle heatmaps show average test accuracy throughout all experiments, with DASH consistently maintaining superior performance across all configurations. Lastly, right heatmaps show average number of steps to converge across all experiments.

### C.4.4 Performance with Continuously Added Data Setting

We designed an experiment mimicking real-world scenarios where new data arrives continuously during training. Following Igl et al. (2020)'s approach, we sampled new data randomly each epoch for the first 500 epochs, combining it with the existing dataset, then continued training for another 500 epochs. This setting assumes a scenario where data continuously arrives before the model has fully converged, as it often does in real-world situations. Cold-started models were trained for 1000 epochs using the entire dataset from the beginning. We used the CIFAR-10 dataset with ResNet18 across five random seeds. We applied both DASH and S&P every 50 epochs through the first 500 epochs with shrinkage parameter $\lambda = 0.3$ for both methods, following a similar setup to Figure 20.

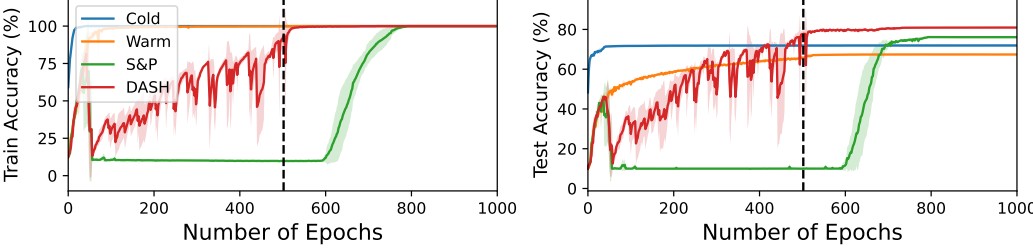

Figure 22: Comparison of training accuracy (left) and test accuracy (right) for ResNet18 on CIFAR-10 over 1000 epochs. During the first 500 epochs, new samples of equal size are added in an i.i.d. manner, followed by 500 epochs of training on the complete dataset. Training accuracy is calculated using only the data available at each epoch during training.

As shown in Figure 22, while there is a clear performance gap between warm-starting and cold-starting approaches, both DASH and S&P aim to address this difference. Despite using the same shrinkage value of 0.3, DASH successfully preserved learned features while S&P did not. This difference is reflected in the training accuracy: DASH showed steady improvement as training progressed, while S&P struggled during the shrinking phases. As a result, DASH not only converged faster but also achieved higher test accuracy, while S&P lost the convergence advantage that warm-starting provides.

### C.4.5 Experiments on Class Incremental Learning Setting

Experiments so far focused exclusively on stationary data distributions. To broaden our understanding, we explored DASH's performance in non-stationary settings, particularly in Class Incremental Learning (CIL) - an important area in continual learning where data distributions shift over time.

We began with experiments on CIFAR-10, dividing the dataset into 10 chunks, each representing a distinct class. The model, based on ResNet-18, was introduced to one chunk at the start of each experiment and trained without access to previously encountered data. However, this setup proved challenging; even with warm-starting as a baseline, the model simply overfitted to each new class, resulting in only 10% test accuracy across all experiments, equivalent to random guessing.

Given these limitations, we adjusted our experimental setup while preserving the non-stationary nature of the task. Using the same configurations with above, instead of completely discarding previous data, we accumulated data over time. Thus, each new chunk was combined with previously seen data. During evaluation, we tested the model only on classes it had encountered during training.

This revised approach yielded more promising results, as shown in Figure 23. DASH surpasses other baselines in terms of test accuracy despite requiring longer convergence times. While these results demonstrate DASH's potential in certain non-stationary environments, we recognize that our modified setup simplifies true non-stationarity, such as in continual learning scenarios.

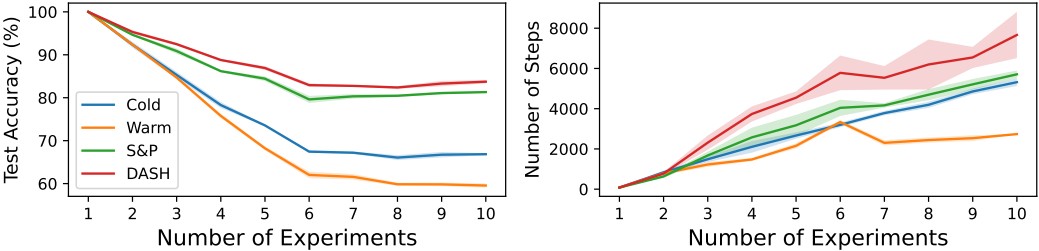

Figure 23: Performance on CIFAR-10 using ResNet-18, averaged over five random seeds. Left: Test accuracy evaluated on classes encountered so far. Right: Number of steps required for convergence.

### C.5 Computation and Memory Overhead Comparison

In this subsection, we will compare the computational and memory overhead of each method. It is important to note that DASH is applied only once when new data is introduced. Since DASH calculates the gradient of the whole dataset just once, its memory complexity is proportional to the batch size and model size. When comparing experiments with the same number of epochs, we can think of DASH as adding approximately one extra epoch to the total training time. Similarly, the computational overhead of DASH is roughly equivalent to running one additional epoch.

We provide the experimental results of the computation and memory overhead for each method in Table 9. The experiments were conducted under the same conditions described in Section 5.1 on CIFAR-10, using ResNet-18 with the SGD optimizer. We provide aggregated results for FLOPS, CPU/CUDA memory usage, and training time on an NVIDIA A6000. These measurements were obtained using the torch.profile library, with values summed across all operations. We only profiled two out of the 50 experiments until convergence (99.9% training accuracy) because profiling adds significant overhead to the training process, and running it continuously slows down the entire training pipeline. A comparison of the total training times of each method for all 50 experiments without profiling is provided in Section C.6. Table 9 indicates that S&P requires two more training epochs than DASH. Since DASH only adds approximately one extra epoch of overhead, the results show that DASH's total computational and memory overhead is about one epoch less than S&P's.

Table 9: Computational and memory requirements for each method, comparing two experiments on CIFAR-10 using ResNet-18. We report the total number of epochs, total training time, and total computational cost (in TeraFLOPS). Memory usage is measured in gigabytes, showing aggregated CPU and CUDA memory consumption.

|  | Epochs | Training Time (s) | TFLOPS | CPU Memory (GB) | CUDA Memory (GB) |
|---|---|---|---|---|---|
| Cold Init | 39 | 29.90 | 21.87 | 7.86 | 813 |
| Warm Init | 30 | 21.96 | 16.82 | 6.03 | 592 |
| S&P | 34 | 25.72 | 19.06 | 6.95 | 690 |
| DASH | 32 | 24.79 | 18.51 | 6.65 | 666 |

## C.6   Comparison between NVIDIA A6000 and Intel Gaudi-v2

We conducted experiments on two different hardware platforms: Intel Gaudi-v2 HPUs and NVIDIA A6000 GPUs. In this section, we compare their implementation code and training times. Using CIFAR-100 and ResNet-18, we measured the training times on both platforms while maintaining the same experimental settings as described in Section 5.1 with SGD optimizer. It is important to note that no hardware-specific optimizations were performed on either platform, and all hyperparameters were kept constant.

To ensure a fair comparison, we ported our NVIDIA implementation to Intel Gaudi-v2, keeping the other settings consistent. The porting process was straightforward, requiring only about three extra lines of code, as shown in the example below. The full Gaudi-v2 implementation can be found in our GitHub repository at https://github.com/NAVER-INTEL-Co-Lab/gaudi-dash.

```
# Import modules
import habana_frameworks.torch.core as htcore
... # additional processing code
for inputs, targets in data_loader:
    outputs = model(inputs)
    loss = criterion(outputs, targets)

    loss.backward()
    # Mark the step for the optimization process
    htcore.mark_step()

    optimizer.step()
    # Mark the step again to complete the optimization cycle
    htcore.mark_step()

    optimizer.zero_grad()
    ... # additional processing code
```

Listing 1: Example of Gaudi-v2 code, with additional lines highlighted in red.

In Table 10, we present the results of the training time comparison until convergence. The table shows that the Intel Gaudi-v2 is slightly faster than the NVIDIA A6000 across all initialization methods. For instance, the Intel Gaudi-v2 achieves a speedup of $1.16\times$ over the NVIDIA A6000 with cold initialization.

Table 10: Training time comparison between two different hardware platforms under the same settings. Intel Gaudi-v2 is slightly faster than NVIDIA A6000.

|  | NVIDIA A6000 | Intel Gaudi-v2 |
|---|---|---|
| Cold Init | 94m 28s | 81m 26s |
| Warm Init | 22m 5s | 21m 16s |
| S&P | 52m 23s | 51m 11s |
| DASH | 40m 8s | 37m 13s |

# D Proof of Theorems

This section provides the proof for Theorems 3.4 and 3.6, stated in Section 3, respectively. Before presenting the main proof, we state some technical lemmas.

**Lemma D.1.** *For any learned feature set $\mathcal{A}, \mathcal{B} \subset \mathcal{S}$ that satisfies $\mathcal{A} \subsetneq \mathcal{B}$ and $|\mathcal{A} \cap \mathcal{S}_c| \geq \tau - 1$ for any $c \in [C]$, then we have $\mathrm{ACC}(\mathcal{A}) < \mathrm{ACC}(\mathcal{B})$.*

*Proof of Lemma D.1.* Since $\mathcal{A} \subsetneq \mathcal{B}$, it is trivial that for any $c \in [C]$ and $\Lambda \subset \mathcal{S}_c$, we have

$$\mathbb{1}\left(|\Lambda \cap \mathcal{A}| < \tau\right) \geq \mathbb{1}\left(|\Lambda \cap \mathcal{B}| < \tau\right). \tag{1}$$

From the given condition, we can choose $c^* \in [C]$ such that there exists $\tau - 1$ distinct features $v_1, \ldots, v_{\tau-1} \in \mathcal{A} \cap \mathcal{S}_{c^*}$ and $v_\tau \in (\mathcal{B} \cap \mathcal{S}_{c^*}) \setminus (\mathcal{A} \cap \mathcal{S}_{c^*})$. Our choice of $\Lambda^* \triangleq \{v_1, \ldots, v_\tau\} \subset \mathcal{S}_{c^*}$ satisfies

$$\mathbb{1}\left(|\Lambda^* \cap \mathcal{A}| < \tau\right) > \mathbb{1}\left(|\Lambda^* \cap \mathcal{B}| < \tau\right). \tag{2}$$

From (1), (2), and the definition of $\mathrm{ACC}(\cdot)$, we have

$$\mathrm{ACC}(\mathcal{A}) = 1 - \frac{C-1}{C} \cdot \frac{1}{n} \sum_{c \in [C], \Lambda \subset \mathcal{S}_c} n_\Lambda \cdot \mathbb{1}\left(|\Lambda \cap \mathcal{A}| < \tau\right)$$

$$< 1 - \frac{C-1}{C} \cdot \frac{1}{n} \sum_{c \in [C], \Lambda \subset \mathcal{S}_c} n_\Lambda \cdot \mathbb{1}\left(|\Lambda \cap \mathcal{B}| < \tau\right)$$

$$= \mathrm{ACC}(\mathcal{B}).$$

$\square$

For ease of presentation, let us say "a model learns `NA`" if a model cannot learn any features. For example, if a model learns $u_1, \cdots, u_s \in \mathcal{S}$ during $s$ steps of training process and feature learning process ends in $(s+1)$-th step, let us say that we learn $u_1, \cdots, u_s, \mathtt{NA}, \mathtt{NA}, \cdots$.

Using the notion above, we prove that our learning process uniquely determines the behavior within the same class regardless of the randomness of the training process, where the randomness may come from tie-breaking that can happen in the choice of the most frequent non-learned feature.

**Lemma D.2.** *Suppose we train two models with different randomness on $\mathcal{T}_{1:j}$ for some $j \in \mathbb{N}$ starting from a learned set $\mathcal{L}$ and without any memorized data. We use $u_s$ and $u'_s$ to denote features learned in $s$-th step of training process by two models, respectively. The $i$-th learned feature within class $c \in [C]$ is denoted as $u_{c,i}$ for the first model and $u'_{c,i}$ for the second model. Then, $u_{c,i} = u'_{c,i}$ for all $c \in [C]$ and $i \in \mathbb{N}$.*

*Proof of Lemma D.2.* Suppose there exists some class $c \in [C]$ and $i \in \mathbb{N}$ such that $u_{c,i} \neq u'_{c,i}$ and choose one with the smallest $i$. Without loss of generality, we may assume $u'_{c,i} \neq \mathtt{NA}$. Then, we have

$$\max_{v \in \mathcal{S}_c \setminus \{u_{c,1}, \ldots, u_{c,i-1}\}} h(v; \{u_{c,1}, \ldots, u_{c,i-1}\}) = \max_{v \in \mathcal{S}_c \setminus \{u'_{c,1}, \ldots, u'_{c,i-1}\}} h(v; \{u'_{c,1}, \ldots, u'_{c,i-1}\})$$

$$= h(u'_{c,i}; \{u'_{c,1}, \ldots, u'_{c,i-1}\})$$

$$\geq \frac{\gamma}{jn}$$

The first equality holds since $\{u_{c,1}, \ldots, u_{c,i-1}\} = \{u'_{c,1}, \ldots, u'_{c,i-1}\}$ from our choice of $c, i$ and the second equality holds since the second model learns $u'_{c,i}$. The last inequality holds since $u'_{c,i} \neq \mathtt{NA}$. Hence, $u_{c,i} \neq \mathtt{NA}$ and

$$h(u_{c,i}; \{u_{c,1}, \ldots, u_{c,i-1}\}) = \max_{v \in \mathcal{S}_c \setminus \{u_{c,1}, \ldots, u_{c,i-1}\}} h(v; \{u_{c,1}, \ldots, u_{c,i-1}\}).$$

From our Assumption 3.3 and since $\{u_{c,1}, \ldots, u_{c,i-1}\} = \{u'_{c,1}, \ldots, u'_{c,i-1}\}$, we have $u_{c,i} = u'_{c,i}$. This is contradictory and we have our desired conclusion. $\square$

**Lemma D.3.** *Suppose we train two models on $\mathcal{T}_{1:j_1}$ and $\mathcal{T}_{1:j_2}$ for some $j_1 > j_2$ starting from a learned set $\mathcal{L}$ and without any memorized data. We use $u_s$ and $u'_s$ to denote features learned in $s$-th step of the training process by two models trained on $\mathcal{T}_{1:j_1}$ and $\mathcal{T}_{1:j_2}$, respectively. The $i$-th learned feature within class $c \in [C]$ is denoted as $u_{c,i}$ for the first model and $u'_{c,i}$ for the second model. Then, $u_{c,i} = u'_{c,i}$ or $u'_{c,i} = \mathtt{NA}$ for all $c \in [C]$ and $i \in \mathbb{N}$.*

*Proof of Lemma D.3.* Suppose there exists some class $c \in [C]$ and $i \in \mathbb{N}$ such that $u_{c,i} \neq u'_{c,i}$ and $u'_{c,i} \neq \mathtt{NA}$. Choose one with the smallest $i$. Since $u'_{c,i} \neq \mathtt{NA}$ and from our choice of $c$ and $i$, we have

$$
\max_{v \in \mathcal{S}_c \setminus \{u_{c,1}, \ldots, u_{c,i-1}\}} h(v; \{u_{c,1}, \ldots, u_{c,i-1}\}) = \max_{v \in \mathcal{S}_c \setminus \{u'_{c,1}, \ldots, u'_{c,i-1}\}} h(v; \{u'_{c,1}, \ldots, u'_{c,i-1}\})
$$
$$
= h(u'_{c,i}; \{u'_{c,1}, \ldots, u'_{c,i-1}\})
$$
$$
\geq \frac{\gamma}{j_2 n} > \frac{\gamma}{j_1 n}.
$$

Hence, $u_{c,i} \neq \mathtt{NA}$ and

$$
h(u_{c,i}; \{u_{c,1}, \ldots, u_{c,i-1}\}) = \max_{v \in \mathcal{S}_c \setminus \{u_{c,1}, \ldots, u_{c,i-1}\}} h(v; \{u_{c,1}, \ldots, u_{c,i-1}\}).
$$

From our Assumption 3.3 and since $\{u_{c,1}, \ldots, u_{c,i-1}\} = \{u'_{c,1}, \ldots, u'_{c,i-1}\}$, we have $u_{c,i} = u'_{c,i}$. This is contradictory and we have our desired conclusion. $\qquad\square$

With above Lemma D.1, D.2 and D.3, we have the following theorems.

**Theorem 3.4.** *There exists nonempty $\mathcal{G} \subsetneq \mathcal{S}$ such that we always obtain $\mathcal{L}_{\mathrm{warm}}^{(1)} = \mathcal{L}_{\mathrm{cold}}^{(1)} = \mathcal{G}$. For all $J \geq 2$, the following inequalities hold:*

$$
\mathrm{ACC}\left(\mathcal{L}_{\mathrm{warm}}^{(J)}\right) \leq \mathrm{ACC}\left(\mathcal{L}_{\mathrm{cold}}^{(J)}\right), \quad T_{\mathrm{warm}}^{(J)} < T_{\mathrm{cold}}^{(J)}
$$

*Furthermore, $\mathrm{ACC}(\mathcal{L}_{\mathrm{warm}}^{(J)}) < \mathrm{ACC}(\mathcal{L}_{\mathrm{cold}}^{(J)})$ holds when $J > \frac{\gamma}{\delta n}$ where $\delta \triangleq \max_{v \in \mathcal{S} \setminus \mathcal{G}} h(v; \mathcal{G}) > 0$.*

*Proof of Theorem 3.4.* By Lemma D.2 for the case $j = 1$, we immediately have our first conclusion by defining $\mathcal{G}$ as a learned feature set from the first experiment. Furthermore, we have $\mathcal{G} \subsetneq \mathcal{S}_c$ and $|\mathcal{G} \cap \mathcal{S}_c| \geq \tau - 1$ for any class $c \in [C]$ from our feature learning framework and Assumption 3.3.

We want to show that for any $J \geq 2$, $\mathcal{L}_{\mathrm{warm}}^{(J)} = \mathcal{G}$. Since we never forget the learned feature in warm training, it is clear that $\mathcal{G} = \mathcal{L}_{\mathrm{warm}}^{(1)} \subset \mathcal{L}_{\mathrm{warm}}^{(J)}$. We may assume that the existence of $J^* \geq 2$ such that $\mathcal{L}_{\mathrm{warm}}^{(1)} \subsetneq \mathcal{L}_{\mathrm{warm}}^{(J^*)}$ and choose the smallest $J^* \geq 2$. Then, in the first step of $J^*$-th experiment, a model learns some feature $u$. From our training process, $u$ satisfies

$$
n \cdot h(u; \mathcal{G}) = |\mathcal{T}_{1:J^*}| \cdot g(u; \mathcal{T}_{1:J^*}, \mathcal{N}_{\mathrm{warm}}^{(J^*, 0)}) \geq \gamma,
$$

and since $J^*$ denotes the first experiment that can learn beyond $\mathcal{G}$, $\mathcal{L}_{\mathrm{warm}}^{(J^*-1)} = \mathcal{G}$ and

$$
|\mathcal{T}_{1:J^*-1}| \cdot g(u; \mathcal{T}_{1:J^*-1}, \mathcal{N}_{\mathrm{warm}}^{(J^*-1, 0)}) = n \cdot h(u; \mathcal{G}) \geq \gamma.
$$

It means that $u$ must have been already learned in the $(J^* - 1)$-th experiment and it is contradictory.

Thus, we have $\mathcal{L}_{\mathrm{warm}}^{(J)} = \mathcal{L}_{\mathrm{warm}}^{(1)} = \mathcal{L}_{\mathrm{cold}}^{(1)} \subset \mathcal{L}_{\mathrm{cold}}^{(J)}$ for all $J \geq 2$ and combining with Lemma D.1, we have

$$
\mathrm{ACC}(\mathcal{L}_{\mathrm{warm}}^{(J)}) = \mathrm{ACC}(\mathcal{L}_{\mathrm{warm}}^{(1)}) = \mathrm{ACC}(\mathcal{L}_{\mathrm{cold}}^{(1)}) \leq \mathrm{ACC}(\mathcal{L}_{\mathrm{cold}}^{(J)}).
$$

To show that strict inequality for $J > \frac{\gamma}{\delta n}$, it suffices to show that $\mathcal{L}_{\mathrm{cold}}^{(1)} \subsetneq \mathcal{L}_{\mathrm{cold}}^{(J)}$ for $J > \frac{\gamma}{\delta n}$ since we already showed that $\mathcal{G} \subsetneq \mathcal{S}$ and $|\mathcal{G} \cap \mathcal{S}_c| \geq \tau - 1$ for any class $c \in [C]$. In $J$-th experiment using cold-starting, by Lemma D.3, a model first learns features in $\mathcal{G}$, say, in the first $s$ steps. In the $(s+1)$-th step, cold-starting model learns a new feature since

$$
\max_{v \in \mathcal{S} \setminus \mathcal{G}} |\mathcal{T}_{1:J}| \cdot g(v; \mathcal{T}_{1:J}, \mathcal{N}_{\mathrm{cold}}^{(J, s)}) = \max_{v \in \mathcal{S} \setminus \mathcal{G}} Jn \cdot h(v; \mathcal{G}) > \gamma,
$$

from the condition in the theorem statement. Hence, we have our conclusion for the test accuracy.

For the train time, since the following holds, we conclude $T_{\text{warm}}^{(J)} < T_{\text{cold}}^{(J)}$ when $J \geq 2$:

$$\sum_{j \in [J]} \left| \mathcal{N}_{\text{warm}}^{(j,0)} \right| = T_{\text{warm}}^{(J)} \leq Jn < \frac{nJ(J+1)}{2} = \sum_{j \in [J]} \left| \mathcal{N}_{\text{cold}}^{(j,0)} \right| = T_{\text{cold}}^{(J)}.$$

□

**Theorem 3.6.** *For any experiment $J \geq 2$, the following holds:*

$$\text{ACC}\left(\mathcal{L}_{\text{cold}}^{(J)}\right) = \text{ACC}\left(\mathcal{L}_{\text{ideal}}^{(J)}\right), \quad T_{\text{warm}}^{(J)} < T_{\text{ideal}}^{(J)} < T_{\text{cold}}^{(J)}$$

*Proof of Theorem 3.6.* Recall that the ideal algorithm works by forgetting memorized data points while retaining previously learned features. In other words, at the initial step of the $(j+1)$-th experiment, we have $\mathcal{L}_{\text{ideal}}^{(j+1,0)} = \mathcal{L}_{\text{ideal}}^{(j)}$. Additionally, $g(v; \mathcal{T}_{1:j+1}, \mathcal{N}_{\text{ideal}}^{(j+1,0)}) = h(v; \mathcal{L}_{\text{ideal}}^{(j+1,0)})$ holds for all $v \in \mathcal{S}$ since $\mathcal{M}^{(j+1,0)} = \emptyset$.

We will show that $\mathcal{L}_{\text{cold}}^{(J)} = \mathcal{L}_{\text{ideal}}^{(J)}$ holds for all $J \geq 1$ by using induction.

When $J = 1$, by applying Lemma D.2, it holds since $\mathcal{L}_{\text{cold}}^{(1)} = \mathcal{L}_{\text{ideal}}^{(1)}$. Suppose $\mathcal{L}_{\text{cold}}^{(J-1)} = \mathcal{L}_{\text{ideal}}^{(J-1)}$ for some $J \geq 2$ and we will prove that $\mathcal{L}_{\text{cold}}^{(J)} = \mathcal{L}_{\text{ideal}}^{(J)}$. We have the following at the first step of the $J$-th experiment for all $v \in \mathcal{S}$:

$$g(v; \mathcal{T}_{1:J}, \mathcal{N}_{\text{ideal}}^{(J,0)}) = h(v; \mathcal{L}_{\text{ideal}}^{(J-1)})$$

For the cold-starting method in the $J$-th experiment, by Lemma D.2, let $s$ be the step at which the model first finishes learning features in $\mathcal{L}_{\text{cold}}^{(J-1)}$. Then, at the $(s+1)$-th step for all $v \in \mathcal{S}$:

$$g(v; \mathcal{T}_{1:J}, \mathcal{N}_{\text{cold}}^{(J,s)}) = h(v; \mathcal{L}_{\text{cold}}^{(J-1)})$$

Since we assumed $h(v; \mathcal{L}_{\text{cold}}^{(J-1)}) = h(v; \mathcal{L}_{\text{ideal}}^{(J-1)})$, the cold-starting method starts to behave identically to the ideal method from the $(s+1)$-th time step onwards, by Lemma D.2, resulting in $\mathcal{L}_{\text{cold}}^{(J)} = \mathcal{L}_{\text{ideal}}^{(J)}$.

$\left| \mathcal{N}_{\text{ideal}}^{(J,0)} \right| < |\mathcal{T}_{1:J}|$ for $J \geq 1$ since $\left| \mathcal{L}_{\text{ideal}}^{(J)} \cap \mathcal{S}_c \right| \geq \tau$ for some class $c \in [C]$ due to Assumption 3.3. Thus, the training time of the ideal method, $T_{\text{ideal}}^{(J)}$, is as follows:

$$\sum_{j \in [J]} \left| \mathcal{N}_{\text{ideal}}^{(j,0)} \right| = T_{\text{ideal}}^{(J)} < T_{\text{cold}}^{(J)} = \sum_{j \in [J]} \left| \mathcal{N}_{\text{cold}}^{(j,0)} \right| = \frac{nJ(J+1)}{2}$$

□

# E   Omitted Algorithms

In this section, we provide detailed training algorithms for our proposed learning framework. Algorithm 2 outlines the standard training method within our learning framework. Subsequently, we compare the Cold-starting, Warm-starting, and Ideal methods using the given abstract algorithm in the following algorithms.

---

**Algorithm 2** Training Process

---

**Require:**
- $\mathcal{L}$: Set of learned features
- $\mathcal{M}$: Set of memorized data points
- $\mathcal{T}$: Training dataset
- $\gamma$: Threshold for learning features
- $\tau$: Threshold for the number of learned features a data point needs to be considered well-classified

1: **function** TRAININGPROCESS($\mathcal{L}, \mathcal{M}, \mathcal{T}, \gamma, \tau$)
2:     **Initialize:**
      $\mathcal{N} \leftarrow \{(\boldsymbol{x}, y) \in \mathcal{T} : |\mathcal{V}(\boldsymbol{x}) \cap \mathcal{L}| < \tau \wedge (\boldsymbol{x}, y) \notin \mathcal{M}\}$
      $s \leftarrow 0$
3:     **while** $\mathcal{N} \neq \emptyset$ **do**
4:         $s \leftarrow s + 1$
5:         $g(v; \mathcal{N}) \leftarrow \frac{1}{|\mathcal{T}|} \sum_{(\boldsymbol{x}, y) \in \mathcal{N}} \mathbb{1}(v \in \mathcal{V}(\boldsymbol{x}))$ for $v \in \mathcal{S}$
6:         $v_s \leftarrow \arg\max_{u \in \mathcal{S} \setminus \mathcal{L}} g(u; \mathcal{N})$   break ties arbitrarily
7:         **if** $g(v_s; \mathcal{N}) \geq \gamma / |\mathcal{T}|$ **then**
8:             $\mathcal{L} \leftarrow \mathcal{L} \cup \{v_s\}$
9:             $\mathcal{N} \leftarrow \{(\boldsymbol{x}, y) \in \mathcal{N} : |\mathcal{V}(\boldsymbol{x}) \cap \mathcal{L}| < \tau\}$
10:        **else**
11:            $\mathcal{M} \leftarrow \mathcal{M} \cup \{(\boldsymbol{x}, y) \in \mathcal{N} : |\mathcal{V}(\boldsymbol{x}) \cap \mathcal{L}| < \tau\}$
12:            $\mathcal{N} \leftarrow \emptyset$
13:        **end if**
14:     **end while**
15:     **return** $\mathcal{L}, \mathcal{M}$
16: **end function**

---

**Algorithm 3** Cold-Starting until $J$-th Experiment

---

**Require:**
- $\mathcal{T}_{1:J}$: Training dataset
- $\gamma$: Threshold for learning features
- $\tau$: Threshold for the number of learned features a data point needs to be considered well-classified

1: **Initialize:**
    $\mathcal{L}^{(0)} \leftarrow \emptyset$
    $\mathcal{M}^{(0)} \leftarrow \emptyset$
2: **for** $j$ **in** $1 : J$ **do**
3:     $\mathcal{L}^{(j)}, \mathcal{M}^{(j)} \leftarrow$ TrainingProcess($\mathcal{L}^{(j-1)}, \mathcal{M}^{(j-1)}, \mathcal{T}_{1:j}, \gamma, \tau$)
4:     $\mathcal{L}^{(j)} \leftarrow \emptyset$
5:     $\mathcal{M}^{(j)} \leftarrow \emptyset$
6: **end for**
7: **return** $\mathcal{L}^{(j)}$

---

---

**Algorithm 4** Warm-Starting until $J$-th Experiment

---

**Require:**

- $\mathcal{T}_{1:J}$: Training dataset
- $\gamma$: Threshold for learning features
- $\tau$: Threshold for the number of learned features a data point needs to be considered well-classified

1: **Initialize:**
$\qquad \mathcal{L}^{(0)} \leftarrow \emptyset$
$\qquad \mathcal{M}^{(0)} \leftarrow \emptyset$
2: **for** $j$ **in** $1:J$ **do**
3: $\qquad \mathcal{L}^{(j)}, \mathcal{M}^{(j)} \leftarrow \text{TrainingProcess}(\mathcal{L}^{(j-1)}, \mathcal{M}^{(j-1)}, \mathcal{T}_{1:j}, \gamma, \tau)$
4: **end for**
5: **return** $\mathcal{L}^{(j)}$

---

---

**Algorithm 5** Ideal-Starting until $J$-th Experiment

---

**Require:**

- $\mathcal{T}_{1:J}$: Training dataset
- $\gamma$: Threshold for learning features
- $\tau$: Threshold for the number of learned features a data point needs to be considered well-classified

1: **Initialize:**
$\qquad \mathcal{L}^{(0)} \leftarrow \emptyset$
$\qquad \mathcal{M}^{(0)} \leftarrow \emptyset$
2: **for** $j$ **in** $1:J$ **do**
3: $\qquad \mathcal{L}^{(j)}, \mathcal{M}^{(j)} \leftarrow \text{TrainingProcess}(\mathcal{L}^{(j-1)}, \mathcal{M}^{(j-1)}, \mathcal{T}_{1:j}, \gamma, \tau)$
4: $\qquad \mathcal{M}^{(j)} \leftarrow \emptyset$
5: **end for**
6: **return** $\mathcal{L}^{(j)}$

---

