# OpenReview forum: "DASH: Warm-Starting Neural Network Training in Stationary Settings without Loss of Plasticity"
_NeurIPS.cc/2024/Conference — NeurIPS 2024 poster_

### Official Review · Reviewer_rFKW · 2024-07-11

**Soundness:** 2
**Presentation:** 2
**Contribution:** 2
**Rating:** 5
**Confidence:** 3

**Summary:**

This paper examines a setting in which a model is learned with an expanding dataset, such that new data is introduced after the model hits a training accuracy threshold. In particular, the new data comes from the same distribution as the existing data (or sometimes has the exact same statistics), making this a stationary learning problem (unlike the non-stationary problems commonly explored in plasticity loss settings). After demonstrating that plasticity loss occurs in this setting, the paper proposes a new method, DASH, to tackle the problem. DASH shrinks weights based on the cosine similarity between the weight vector and the negative gradient of the training loss. There are a series of experiments which generally show DASH outperforming other plasticity loss prevention approaches, with more in the appendix.

**Strengths:**

- Based on my knowledge and research, I think this is a novel approach to preventing plasticity loss. It is quite original in its idea, which the authors have tried to ground in theory.
- This is a relatively underexplored area (plasticity loss under stationary settings), and thus is useful for understanding plasticity loss further.
- DASH seems to offer genuine performance improvements over most methods, though it is worth noting that this is not always the case (i.e., in the appendix more results are included which are either significantly more marginal, or sometimes favouring other methods). In particular, the focus that test accuracy is not only improved, but we carry over some of the benefits of warm starting (i.e., quicker convergence) is notable.
- The figures/graphs in the paper were generally good and useful for understanding what was being discussed.

**Weaknesses:**

Before noting weaknesses of the paper, which I generally feel come down to clarity issues. I am open to increasing my score in light of discussion based on the author's response to my highlighted weaknesses and questions.

- I found a lot of the theory quite confusing, and couldn't follow section 3 ('Comparison Between Warm-Start and Cold-Start'). Similarly, I found some explanations about the method, and the training process quite confusing. Perhaps demonstrating with examples could be useful, but it is currently quite hard to parse exactly what the problem setting is.
- This work has not been studied in a particularly standard learning process. Instead, the focus is on a setting where features are learned based on their frequency of appearance in data, rather than their signal strength. This is quite an artificial setting, and it is not exactly clear to me that DASH scales to more conventional learning frameworks.
- Appendix A.3 feels slightly burrowed away, but considers the SOTA setting for CIFAR-10 and CIFAR-100, in which DASH underperforms random initialisation. I think this is quite a crucial result, and arguably outweights the results from Table 1 in which a more naive approach is taken. While it is good that the authors include these results in the appendix, it would be good to try to include this in a more upfront way (perhaps including discussion about how DASH is helpful in a naive setting and helps reduce the barrier to entry with less studied datasets?)

**Questions:**

- Considering a different learning setting, where data is continuously added rather than in predictable batches, would DASH still be able to perform (given at the moment, it is applied whenever new data is provided)? Similarly, how is its performance when we are not adding data from a stationary distribution, given this is possibly more realistic?
- How do the authors think DASH could work in the more generic learning problem (i.e., where we are not learning features based on their frequency)?
- What is actually meant by the frequency of features - can you offer an example please? I find it hard to determine exactly what is meant by the frequency of these discrete features?

- In table 8, the authors write CIF1R rather than CIFAR.
- What is meant in line 325: 'This allows weights to easily change its direction.'?

**Limitations:**

- In the appendix, there are examples where DASH does not outperform other methods, or is within the error margin. However, there is clearly a strong signal that DASH on average provides benefit to the training in all but the SOTA setting. The authors are upfront about this.
- I believe the authors are generally upfront about the limitations of their work, though I feel perhaps highlighting that they are working in a slightly artificial setting, and demonstrating how DASH could be implemented to a typical machine learning pipeline, would be useful.

---

> ### Author Rebuttal · Authors · 2024-08-07
>
> We sincerely appreciate the reviewer for providing valuable and constructive feedback on our work. We also thank you for bringing to our attention typos in Table 8. Below, we address the questions and concerns raised by the reviewer.
> ## **W1, Q3. Clarification on our Framework and DASH**
> We acknowledge that there may be some confusion regarding our theoretical framework. To clarify, we provide a high-level intuition and an example to the global response to better explain our framework. We hope this will help in understanding our approach. In our next version, we will include these explanations.
>
> Regarding Q3, in our framework, the frequency of a feature in dataset is estimated by the number of data that contain the feature, divided by the total number of data. For example, in Figure 1 of the attached pdf, the feature represented by the violet has a frequency of 5/6. Also, we emphasize that the concept of learning by the frequency of features in our framework is intended to capture the characteristics of learning in real-world in abstract manner.
>
> We hope this clarifies any confusion.
> ## **W2, Q2, L2. DASH is practical method**
> We believe that there may be some confusion. DASH is a practical algorithm that has been evaluated on real-world datasets such as CIFAR-10, CIFAR-100, and Tiny ImageNet. We suspect that we didn’t deliver this well enough and made the reviewer think that DASH operates only within the feature learning framework introduced in Section 2. However, that framework is intended to mathematically model the training procedure and hence analyze the test accuracy and training time trade-off between cold-start and warm-start. We propose a method which is ideal in our framework and show that the method mitigates the trade-off. Motivated by this ideal method, we propose DASH as a practical method which intuitively captures the idea of the idealized method.
> ## **Q5. What is meant 'This allows weights to easily change its direction.'?**
> It means that the mechanism enables weights to change their direction with even small updates. Here’s why this is important:
>
> In unseen data, features are more likely to reappear, while noise from old data is less likely to be present. As a result, learning these features effectively enhances generalization. In other words, it is beneficial for neurons to align with class-relevant feature directions. By shrinking weights that are less aligned with these features, we make it easier for the model to adjust and align with the feature directions and enhance generalization performance.
> ## **W3 & L1 Underperformance DASH in SOTA setting**
> We agree that there was a lack of discussion in the main text regarding the underperformance of DASH compared to cold-starting in the SOTA setting, which we had only addressed in the appendix. We plan to include this in the main text in the next revision. As described in Appendix A.3, we believe that this underperformance is partially due to using hyperparameter optimized for cold-starting. Even though we did not tune the hyperparameters for DASH, it remains comparable to cold-start, which in fact better aligns with our theory (Theorem 4.1.). We believe that specifically tuning the hyperparameters for DASH might further enhance its performance and potentially surpass the cold-start method in the SOTA setting. The fact that our method performs better with naive training suggests it might be particularly effective for understudied datasets where SOTA settings are not well-established, as you pointed out.
> ## **Q1. The setting in which data is continuously added**
> While it is not entirely clear to us what the reviewer meant by “continuously added data”, we conducted an additional experiment that might address your concern. In this experiment, we added new data at each epoch for the first 500 epochs, then continued training for an additional 500 epochs (similar to Igl et al. (2021)). Cold-started models are trained for 1000 epochs using the entire dataset from the beginning. We used the CIFAR-10 on ResNet-18 with five random seeds.
>
> As shown in the result plot in Figure 7 in attached PDF, there is a clear performance gap between warm-starting and cold-starting methods. We applied DASH and S&P every 50 epochs periodically until the 500th epoch, as you mentioned it cannot be done every time new data is introduced. Interestingly, DASH preserved learned features while S&P did not, even when we used the same $\lambda$ value of 0.3. As a result, DASH showed an increasing trend in train accuracy as the number of epochs increased, while S&P failed to learn anything during shrinking. Consequently, DASH converged faster than S&P while achieving better test accuracy.
> ## **Q1.  The setting in which data is added from non-stationary distribution**
> DASH is designed for stationary data distributions, assuming features of the new data remain consistent with previous data. Because of this, the negative gradient has a high cosine similarity with the neurons that already learned features, allowing learned features to be retained. However, in non-stationary cases, the features may change continuously as new data is introduced. This means that it might be challenging to retain already learned features in such scenarios. The effectiveness of DASH may be limited in these non-stationary environments.
>
> Adapting DASH for non-stationary scenarios (e.g., reinforcement learning) might require incorporating techniques similar to S&P, where perturbations are introduced to reactivate dead neurons. This approach could be crucial for maintaining network adaptability in changing data distributions. However, addressing these challenges in non-stationary environments is beyond the scope of our current work. We believe that this is a very important future direction.
>
> Thanks for your time and consideration.
>
> Best regards,
>
> Authors
>
> ---
> Reference
> - Igl et al. Transient Non-Stationarity and Generalization in Deep Reinforcement Learning. In ICLR 2021

---

> ### Author Response · Authors · 2024-08-12
>
> Dear Reviewer rFKW,
>
> We sincerely appreciate your time in reviewing our work. Your feedback has helped us improve our research. We understand that you may have a busy schedule, but we'd be grateful if you could take a moment to look over our responses and ensure we've addressed your concerns adequately. If you have any additional questions or suggestions, please don't hesitate to let us know. We're ready to address any further points you may have.
>
> Thank you again for your valuable contribution to our research.

---

> > ### Comment · Reviewer_rFKW · 2024-08-12
> >
> > Dear authors,
> >
> > Firstly, let me apologise for my delay in responding to your rebuttals. I will attempt to remain more active in the coming days to enable active discussion, if necessary.
> >
> > I still hold a few concerns regarding the performance of DASH:
> > - It is quite difficult to make claims about the significance of DASH given the performance in the SOTA setting. Despite the author's beliefs that this may improve with hyperparameter tuning, I would need to see this run experimentally to believe this is the case (I respect that you did not have the time to run this experiment over the rebuttal period due to the strict time constraints).
> > - I am still not certain I believe in even the intuition of the feature learning framework; as I say, this ignores the fact that different signals provide varying levels of signal for learning, and thus the frequency of features is not the only factor that determines learning.
> >
> > That said, additional results provided in the global response have strengthened your paper in my opinion. I also agree that it would be interesting to see how DASH could be applied to nonstationary settings like RL, and would suggest adding this as a proposal for obvious future work.
> >
> > I have raised my score to recommend borderline acceptance.

---

> ### Author Response · Authors · 2024-08-13
>
> Thank you for your feedback and for reconsidering the score.
>
> - While we currently don't have the time to fine-tune hyperparameters for DASH in the SOTA setting, we intend to address this in our next revision. In the upcoming update, we will perform more comprehensive hyperparameter tuning, tailored specifically to each method, and incorporate these improved results into our findings.
> - Thank you for your additional comments on our feature learning framework. We now understand the concerns you raised in W2 and Q2. As you pointed out, the learning of features can also be influenced by the strength of the features, especially when distinct features vary in strength. Our theoretical results and framework can be directly extended to this scenario by slightly altering our approach to consider $(\text{frequency}) \times \text{(strength)}$ as learning criteria instead of just frequency. Additionally, we believe our analysis can be extended to scenarios where feature strength varies across data by treating the set of features as a multiset, where multiple instances of the same element are allowed. Since the analyses in these cases are nearly identical to the one we have considered and follows the same underlying principles, we have assumed all features have identical strength for notational simplicity, without loss of generality. We hope these discussions adequately adress your concerns. We plan to include these discussions as a remark in the next revision to provide a clearer explanation to readers of our learning framework.
>
> We welcome any further questions or comments you may have.

---

### Official Review · Reviewer_hPM9 · 2024-07-12

**Soundness:** 3
**Presentation:** 3
**Contribution:** 3
**Rating:** 6
**Confidence:** 4

**Summary:**

The paper examines warm-start and the loss of plasticity, identifying that noise memorization from warm-start impairs generalization in stationary data distributions. It proposes a method called DASH, which selectively forgets previously memorized noise while preserving learned features.

**Strengths:**

- **Feature Learning Framework**: The paper introduces a feature learning framework to investigate the warm-start issue. It thoroughly explores the properties of warm-start and cold-start training, the role of noise memorization in the loss of plasticity, and the selective retention (forgetting) of features (noise). These core findings are well-supported through theoretical grounding and experimentation.
- **Performance:** In vanilla settings, DASH significantly outperforms other methods on several small datasets in terms of accuracy and training steps. And, in the SoTA setting, DASH rivals other methods.

**Weaknesses:**

- **Compute Overhead**: Gradient-based approaches typically incur significant compute overhead. While DASH is claimed to achieve faster convergence, it remains unclear if its shrinking process slows down the overall training. Quantitative comparisons in terms of FLOPs and training time would be insightful. Furthermore, DASH has not been compared with baselines regarding compute and memory overhead. The training convergence based on the gradient norm does not consider the compute and memory overhead associated with DASH’s gradient calculations.
- **Assumption: Gradient norm as a proxy for the number of training steps**: The assumption that the number of training steps can be reliably approximated by the initial gradient norm is insufficiently supported by the data presented in Figure 10 of the paper. The correlation between training steps and the initial gradient norm does not appear to be linear, particularly for the Warm start.
- **In-depth Analysis**: The experiments do not demonstrate a direct link between learned/memorized data and the cosine similarity of gradients and weights. Conducting ablation studies might provide further insights into this relationship. Additionally, the proposed method appears to be a variant of the shrink and perturb approach.
- **Evaluation and Scalability**: Evaluation is performed exclusively on toy datasets, leaving DASH’s scalability to larger and more challenging datasets, such as ImageNet-1K, unknown.
- **Generality**: In SOTA settings, as shown in Table 7, DASH does not consistently outperform the compared methods, particularly warm initialization. This raises questions about the wider applicability and robustness of DASH. Its efficacy seems to diminish when using augmentations and weight decay.

**Questions:**

- Does the apparent non-linearity in the correlation between initial gradient norms and training steps in Figure 10 change how Figure 1(b) should be interpreted?
- In Figure 2, the test accuracy drops with longer pre-training when the model is pre-trained on 50% of the training data and then fine-tuned on the full dataset. Is this phenomenon universal or does it only occur with the CIFAR dataset? Could the authors provide similar results for high-dimensional datasets, such as ImageNet-1K (224x224)? If the network learns the same set of semantic categories during the pre-training and fine-tuning stages, the fine-tuning stage only differs in terms of the number of samples. Is this phenomenon related to the size of the training data? How much do pre-training and fine-tuning datasets differ in terms of distribution shifts? Is the pre-training data class balanced? How does DASH perform in a similar setting?
- Does DASH need access to the entire dataset (all chunks), including current and previous datasets/chunks, for computing gradients and shrinking weights during every training step?
- In *Line 116*, the set `S_y` is undefined, and may be corrected to `S_c`.
- In *Line 134*, `Tau < K` may be corrected to `Tau <= K`, where the number of learned features can also equal the set of learnable class-dependent features, K.

**Limitations:**

The paper provides little to no detail on the limitations of this work. While it mentions the assumptions of a discrete learning process and the lack of consideration for a hypothesis class, it does not address whether DASH can be extended to larger problem settings. If such limitations exist, they should be explicitly stated by the authors.

---

> ### Author Rebuttal · Authors · 2024-08-07
>
> We sincerely appreciate the reviewer for providing valuable and constructive feedback on our work. Below, we address the questions and concerns raised by the reviewer.
>
> ## **W1 & W3**
> Check out our global response where we address these issues. We hope our explanation and ablation study resolve all concerns you may have.
>
> ## **W2, Q1. Proxy for the number of training steps**
> We believe there may be some confusion, let us clarify this point. For all our real-world experiments including Figure 1(b), we count the actual number of training steps until near-convergence (99.9% training accuracy), not the initial gradient norm. In contrast, we use the number of non-zero gradient data as a proxy of the training time ONLY in our abstract learning framework, for the sole purpose of theoretical analysis.
>
> The reason for adopting a proxy of the training time is to capture the concept of “training time” within our abstract learning framework in which features/noise are learned/memorized in an “on/off” manner. Due to the binary (zero vs non-zero gradient) nature of our framework, we think the number of non-zero gradient data can serve as a reasonable proxy of the norm of gradient in practice. In addition, since the initial gradient norm is positively correlated with the number of training steps (in practice), we believe that the number of non-zero data point in our abstract framework can role as proxy for the number of training steps in practice. While the relationship between gradient norm and training time is not strictly linear, we think the number of non-zero gradient data to be the best possible measure of training time in our framework.
>
> ## **W4, Q2. Evaluation on ImageNet-1K**
> We appreciate your comments on scalability for larger datasets like ImageNet-1K. However, including experimental results for such datasets are challenging, as it would require repeating the training process 50 times until convergence—an extremely time-consuming process.
> As an alternative, we trained ImageNet-1K on ResNet18 using a setup similar to Figure 2 in Section 3.2. We used 50% of the data for pretraining and the full dataset for fine-tuning. Results in Figure 3 (attached PDF), show DASH outperforming both cold and warm starts in test accuracy and convergence speed, demonstrating its effectiveness on challenging datasets. Due to time constraints, we couldn't include results from training ImageNet-1K divided into 50 chunks in the paper. However, we plan to add results in the future using more than two chunks.
>
> We also investigate the effect of pretrain epoch on warm-starting in ImageNet-1K classification with ResNet18. We conducted experiments using different pretrain epochs: 30, 50, 70, and 150. Figure 4 (attached PDF) shows a decline trend in accuracy for warm-started models as pretrain epoch increases.
> Interestingly, we noted a similar phenomenon in Figure 2 in Section 3.2, where warm-starting does not negatively impact test accuracy when training resumes from earlier pretrained epochs, such as 30 in this case. This observation aligns with our theoretical framework, which suggests that neural networks tend to learn meaningful features first and then begin to memorize noise later. We believe this phenomenon is persistent across different datasets, model architectures, and optimizers.
>
> ## **W5. Inconsistency in outperformance of DASH compared to warm-start**
> We acknowledge that there is some case that DASH underperform warm-start in the test accuracy at last experiment. However, It is important to note that the performance differences among all methods are not large, with most falling within standard deviation ranges. Also, we emphasize that performance at every experiment is crucial in practice. Therefore, we suggest examining not only the test accuracy at the final experiment but also the average test accuracy across all experiments. We ask the reviewer to revisit Table 7 in Appendix A.3., where we show that DASH consistently outperforms warm-starting in terms of average test accuracy, with improvements ranging from as little as 0.45% to as much as nearly 3%.
>
> ## **Minor Questions**
> - Q2. You're right that this is a matter of training data size. If a network isn't provided with sufficient data to learn enough features during the initial experiment, it may struggle to learn feature even when new data is introduced at following experiments. This concept is central to our Theorem 3.4. In scenarios where data is sampled from a stationary and class-balanced distribution (like real-world image data), there is less distribution shift due to class imbalance, as the samples are drawn iid.
> - Q3. DASH is applied at the start of each experiment, not at every training step. When DASH's hyperparameter $\alpha$ is set to 1, only new data is needed for shrinking. Figures 7-8 in Appendix A.4 show DASH behaves similarly across different $\alpha$ values, including when $\alpha=1$. This indicates that DASH can be effective even when only the new data chunk is accessible.
> Furthermore, DASH can be utilized in online learning settings, where only newly introduced data is available during training. We evaluate DASH in such a setting using CIFAR-10 divided into 50 chunks and trained on ResNet18. Each experiment was conducted on a single chunk of data before moving to the next, with all other settings consistent with those detailed in Section 5.1. The results, shown in Figure 8 (attached PDF), show that periodically applying DASH outperforms baselines including S&P in terms of test accuracy. This confirms DASH's effectiveness with limited data access.
> - Q4. Our use of $S_y$ is intended to represent the set of features associated with the true label $y$.
> - Q5. We consider the case where $\tau$ is strictly smaller than the total number of features within a class. If $\tau = K$, then only data points containing all possible features would be correctly classified during testing.
>
> Thanks for your time and consideration.
>
> Best regards,
>
> Authors

---

> > ### Comment · Reviewer_hPM9 · 2024-08-09
> >
> > I thank the authors for their detailed responses, clarifications, and additional experiments. Some of my concerns are addressed by the authors.
> >
> > I have a few more questions and concerns. Although the concern regarding online learning is raised by reviewer LBum, I also need clarification regarding this.
> >
> > - Regarding Figure 8 in the attached PDF, could you explain why the test accuracy of DASH and S&P sometimes decreases and is less smooth compared to the "warm" method? Additionally, can you confirm that the model is only trained on new data or the new chunk, rather than on all previously seen chunks?
> >
> > - In your online learning experiments (Figure 8), I assume each chunk contains almost all classes from the CIFAR-10 dataset. I’m interested in understanding whether there is any distribution shift between subsequent chunks. In continual learning, there is a setting called class incremental learning (CIL), where chunks or batches of data are mutually exclusive, meaning classes do not overlap across chunks. For example, if CIFAR-10 is divided into 10 chunks, each chunk would contain only one unique class. If DASH were applied in this setting, how effective would shrinking by cosine similarity be? Could you explain this in terms of learned features and noise?
> >
> > - In contrast, there’s another setting in continual learning called IID (independent and identically distributed), which is the opposite of CIL. In the IID setting, chunks or batches contain overlapping classes, leading to minimal distribution shift, unlike in CIL where the shift is significant. Consequently, catastrophic forgetting is significant in CIL and negligible in IID. I assume all your experimental settings in the paper and rebuttal are based on IID, is that correct?
> >
> > - My primary question is: Is DASH only applicable in scenarios where subsequent chunks/batches/experiments contain the same/ overlapping classes (IID setting)? If so, it cannot be claimed that DASH addresses catastrophic forgetting.
> >
> > It would also be helpful to include S&P in Figure 3 of the attached PDF. I suggest incorporating the additional experiments and results into the paper and clarifying the applicability of DASH in response to my questions.

---

> > > ### Author Response · Authors · 2024-08-10
> > >
> > > ### **3-1. Explanation of DASH on Online CIL Setting**
> > >
> > > We acknowledge that Class Incremental Learning (CIL) is an important research topic in continual learning where data distribution is non-stationary. Following your suggestion, we conducted experiments to explore DASH's effectiveness in a CIL setting, which contrasts with our original setup.
> > >
> > > We divided CIFAR-10 into 10 chunks, each containing data from a single class. In this setting, at the start of each experiment, a chunk with a specific label is introduced, and the model is trained using only this new chunk without access to previous data. We trained CIFAR-10 on ResNet-18 with five random seeds.  Please let us know if you have a different setup in mind.
> > >
> > > We first conducted experiments with warm-starting as a baseline in this setting. The results showed that the model overfits the data introduced in each experiment, predicting only the corresponding label, resulting in a test accuracy of around 10% for all experiments. In this online learning scenario, it seems unlikely that any method manipulating the initialization at the beginning of every experiment, including warm-starting, DASH, or S&P, would be effective. In fact, both DASH and S&P performed no better than random guessing, also achieving only 10% test accuracy, simillar to  warm-starting. Instead, we might need to modify the model architecture or employ other training techniques e.g. replay buffer or regularization.
> > >
> > > Despite this, it's worth considering why DASH performs as bad as warm-starting. The primary reason is the limited overlap between class-dependent features across different data chunks. If we apply DASH at the beginning of each experiment, even neurons that learned features from previous chunks would have low cosine similarity with the negative gradients of the new chunk. This leads to significant shrinking and forgetting. It's important to note that our algorithm was originally designed for stationary settings. As we mentioned in our response to rFKm, non-stationary cases require different considerations. We view this as an important direction for future work.
> > >
> > > ### **3-2. CIL with “Data Accumulation” Setup**
> > >
> > > The setting described above is not suitable for evaluating DASH's applicability, as even warm-starting performs no better than random guessing. Given that even a warm-started neural network struggles in a standard CIL setting, we conducted additional experiments in a modified CIL environment. Similar to the previous setup, we divided CIFAR-10 into 10 chunks, each containing data from a single class. This time, instead of discarding data from previous experiments, we combined newly introduced data with the existing training data (i.e., the same as the “data accumulation” setup considered in our paper). This approach aligns more closely with our main focus while still maintaining the core concept of non-stationarity you mentioned. During testing, we evaluated the model only on the labels it had encountered during training.
> > >
> > > The model trained with warm-starting achieved a final accuracy of $59.55\% (\pm 0.53\%)$ in this setting, with an average test accuracy of $72.39\% (\pm 0.53\%)$ across 10 experiments. In contrast, when applying DASH, we achieved a final accuracy of $83.74\% (\pm 0.45\%)$ with an average test accuracy of $87.86\% (\pm 0.18\%)$.
> > >
> > > These results show that DASH can be effective in certain non-stationary settings, though our modified setup doesn't fully represent a truly non-stationary environment. While we can't entirely explain DASH's strong performance in this context with our abstract theoretical framework (Section 2), we attribute it to two key factors.
> > >
> > > The first factor is our experimental approach. By incorporating data from previous experiments, we allow this information to continue contributing to the learning process. This is why warm-starting tends to yield better test accuracy in this scenario. Another key factor is DASH’s algorithm design. Even when DASH shrinks all weights to their lower bound $\lambda$ because of low cosine similarity, it would likely perform similarly to S&P.
> > >
> > > ### **4. Regarding IID Setting**
> > >
> > > You're correct in noting that we focused on the IID setting. We aim to address why warm-starting underperforms in this IID setting (i.e., the loss of plasticity) and to propose a solution to this issue. Thus, we would like to emphasize that the significance of our research in the IID setting remains important and should not be overlooked.
> > >
> > > ### **5. Regarding S&P in Figure 3 of the attached PDF**
> > >
> > > Due to the computation-intensive nature of the experiments on ImageNet-1K, we couldn’t make a direct comparison with S&P at this time. We appreciate your suggestion and plan to include this comparison in the revised version of our paper.
> > >
> > >
> > > ---
> > >
> > > We hope that our response clarifies your concerns. We would appreciate it if you could reconsider your assessment.

---

> > > ### Author Response · Authors · 2024-08-10
> > >
> > > **References**
> > >
> > > [1] Nikishin et al., The Primacy Bias in Deep Reinforcement Learning, ICML 2022
> > >
> > > [2] Zhou et al., Fortuitous forgetting in connectionist networks, ICLR 2022
> > >
> > > [3] Ash and Adams, On warm-starting neural network training, NeurIPS 2020

---

> > > > ### Comment · Reviewer_hPM9 · 2024-08-11
> > > >
> > > > I appreciate the authors for providing additional clarifications regarding the IID/CIL settings and the forgetting phenomenon, which overlap in continual learning and the shrinking process.
> > > >
> > > > The authors have adequately addressed my major concerns, presenting solid contributions toward resolving warm-starting issues and the loss of plasticity. Therefore, I have raised my rating and believe this paper could be accepted for publication.
> > > >
> > > > I suggest the authors clarify these points in the main paper and release the code to ensure reproducibility and transparency.

---

> > > > > ### Author Response · Authors · 2024-08-11
> > > > >
> > > > > Thank you for your response. We are glad to hear that our explanations were helpful and adequately addressed your concerns. We also appreciate your suggestions for improvement and plan to incorporate them into our next revision.
> > > > >
> > > > > Best regards,
> > > > >
> > > > > Authors

---

> ### Author Response · Authors · 2024-08-10
>
> Thanks for your response. We are glad to hear that some of your concerns have been addressed. We want to respond to your additional comments regarding our supplementary experiment in the online learning setting.
>
> ### **1. Clarification on “Catastrophic Forgetting”**
>
> Before directly answering the questions one at a time, we want to clarify what we meant when discussing the reduction of “catastrophic forgetting” in an IID setting. In our experiment setup, we considered an online learning scenario where all data chunks are drawn from an identical distribution. While catastrophic forgetting is typically less severe in this IID setting as you pointed out, the inability to access previously introduced data in later experiments can still lead to significant forgetting issues. This is particularly problematic with shrinking-based methods, as they intentionally force the model to forget previously learned information through the shrinking process [3].
> We considered the inability to retain learned information as a form of catastrophic forgetting. We apologize for any confusion this may have caused. To clarify, when we referred to "catastrophic forgetting" in our previous response to Reviewer LBum, we meant to highlight that shrinking-based methods inherently face this “forgetting” issue. Our point was that applying DASH and S&P for every chunk doesn't actually solve this problem. Instead, we attempt to mitigate the issues associated with shrinking-based methods by applying shrinking with some intervals.
>
> ### **2. Degradation of DASH and S&P in Figure 8**
>
> Every point at which degradation of test accuracy occurs in DASH and S&P coincides with the shrinking point of each method. The reason for different shrinking cycles is that we report the best learning curve (in terms of the last test accuracy) out of all $3\times3=9$ hyperparameter configurations (the interval between shrinkages $\in \\{10, 15, 20\\}$ & the shrinkage factor $\lambda \in \\{0.05, 0.1, 0.3\\}$) for each method. The performance drops immediately after applying parameter modification (i.e., resetting or shrinking) and subsequent performance recovery is common in the literature of resetting techniques [1,2].
>
> Additionally, in our online learning setting, we only used a newly introduced chunk of data for training at each experiment, rather than training on all previously seen data. This resulted in slower accuracy recovery due to the limited amount of training data. Moreover, when applying shrinking (DASH or S&P) too frequently, the performance degradation caused by shrinking outweighs the learning effect from the limited number of new data points, leading to a decrease in overall test accuracy.

---

### Official Review · Reviewer_t732 · 2024-07-13

**Soundness:** 3
**Presentation:** 3
**Contribution:** 3
**Rating:** 5
**Confidence:** 2

**Summary:**

Warm-starting neural networks may lead to poor generalization performance or loss of plasticity, likely due to overfitting. This paper presents a framework that hypothesizes that noise memorization is the primary cause of loss of generalization in warm-starting settings. The authors then present an algorithm motivated by their framework that improves the generalization and plasticity of the neural networks. The algorithm works on a per-neuron level and measures the cosine similarity between the negative gradient of the input weight to each neuron and its weights. This cosine similarity is used as a utility measure to allow for more shrinking for the input weights with low cosine similarity. The authors showed the effectiveness of their method on a wide range of problems and compared it against other baselines demonstrating superior performance.

**Strengths:**

The paper addresses an important problem: maintaining plasticity and generalizing neural networks. The proposed algorithm is novel and effective, and the experimental results seem comprehensive.

**Weaknesses:**

- The main problem of this work is that the link between the framework and the algorithm is not clear. For example, it’s unclear why different parts of the algorithm are designed this way, especially the cosine similarity part. Why is the momentum direction assumed to be the direction of features?
- The evaluation may be unfair and statistically insignificant. In multiple experiments (e.g., figure 1),  the results are generated based on three seeds. This is a very low number to have statistically significant results. Additionally, the reported methods use the same hyperparameter across multiple methods, which may be unfair to the baselines since they are not well-tuned compared to DASH. The results should be based on the best hyperparameter of each method. This might explain why the authors found approaches such as L2 Init, resetting, layer normalization, SAM, or reviving dead neurons to be less effective. In contrast, the only effective one was S&P, which uses a hyperparameter (shrinking) similar to the DASH method.

**Questions:**

- In Figure 4, the authors presented an arrow with a feature direction. What does that mean exactly in a rigorous way?
- What is the definition of a feature? The word is used in the paper many times without a proper definition, leading to further confusion. In neural networks, the learner always improves its set of features, so they are always changing.

**Limitations:**

No. I think the main limitation of the method is that the cosine-similarity metric seems to be heuristic-based and not motivated by the introduced framework. It is unclear if such a metric is ideal or if it reflects what is suggested by the theoretical framework.

---

> ### Author Rebuttal · Authors · 2024-08-07
>
> We would like to express our appreciation to the reviewer for your valuable and constructive comments.  In the following, we address the points raised by the reviewer.
>
> ## **W1, Limitation. Clarification on our framework and link between the framework and the algorithm**
>
> We apologize for the unclear explanation regarding the framework and connection to our algorithm, DASH. We have clarified this in the general response. In it, we explain the role of cosine similarity and how shrinking with cosine similarity effectively captures feature retaining and noise forgetting. We hope this addresses any concerns.
>
> ## **W2. Additional results with different seeds**
>
> We understand your concerns about the statistical significance of our experimental results. While we aim to include as many random seeds as possible, our incremental learning framework is computationally intensive, which initially limited us to three random seeds. To address this issue, we're actively working on increasing the number of random seeds to enhance statistical significance.
> In response to your concerns, we can now provide results from two additional seeds (bringing the total to five) for experiments trained with SGD. As shown in Table 1 attached PDF in the global response, these new results demonstrate trends similar to our previously reported values. We hope this improvement addresses part of the concerns you raised.
>
> Unfortunately, we cannot currently provide results for Tiny-ImageNet on VGG16 with SGD, as these experiments take approximately a week to train. However, we'd like to draw your attention to the results table in Appendix A.2, which shows our approach with Tiny-ImageNet on VGG outperforming other baselines. We hope this helps alleviate some of your concerns. In our revised version, we plan to include results with five random seeds for all experiments to ensure comprehensive statistical significance across our findings.
>
> ## **W2. Additional results with different hyperparameters**
>
> In our experiments, we used the same hyperparameters across multiple datasets and methods. It's important to note that these hyperparameters were not specifically tuned for DASH; rather, this approach was chosen for fair comparison, following the methodology of Ash & Adams 2020.
> Ideally, as you pointed out, we would have preferred to use the best hyperparameters for each method. However, due to limited computational resources, we couldn't exhaustively search for optimal values of base hyperparameters (such as learning rate and batch size) or method-specific parameters, since some runs take as long as a week. Given these constraints, we concluded that the best way to compare baselines was to maintain consistent hyperparameters across all methods.
>
> In our original vanilla setting described in Section 5.1, the only hyperparameters available for tuning were learning rate and batch size. To address your concerns, we conducted additional experiments with CIFAR-10 on ResNet18 where we varied the learning rate while keeping the batch size fixed at 128.
> You can see these results illustrated in Figure 5, 6 and Table 2. Our findings confirm what we previously mentioned: there is little to no effect when using L2 Init, resetting, layer normalization, SAM, or reviving dead neurons. L2 Init and resetting show worse generalization performance compared to warm-started neural networks.
>
> ## **Questions**
>
> We have provided high-level intuition and explanations with examples for our framework in our global response, and we encourage you to refer to it. We hope this will clarify any misunderstandings.
>
> **Q1. What does the feature direction mean exactly in a rigorous way?**
>
> As detailed in our global response, there are two ways to minimize training loss: learning features and memorizing noise, with only learning features being beneficial for unseen test data. Therefore, the negative gradient can be decomposed into components of feature and noise. A feature direction represents *the direction of updates that is effective for both training and test data*.
>
> **Q2. What is the definition of feature?**
>
> We emphasize that the notion of features in our work is totally different from what is commonly referred to as 'features' in the context of neural network outputs, specifically the last hidden layer outputs. As discussed in our global response, we consider features to be the information contained in the input that is relevant to the label and therefore useful for generalizing to unseen data.
>
> Thanks for your time and consideration.
>
> Best regards,
>
> Authors
>
> ---
> Reference
>
> - Jordan Ash and Ryan P Adams. On warm-starting neural network training. In NeurIPS 2020

---

> > ### Comment · Reviewer_t732 · 2024-08-12
> > **Thank you for your response**
> >
> > I thank the authors for their detailed response. After reading the response and other reviews, I think my initial assessment has been confirmed. I think major restructuring is needed to build a logical flow between the framework and the algorithm. While the authors explained the intuition, it still lacks some rigor, and judging a completely rewritten introduction and method sections requires another review process. Therefore, I maintain my original score.

---

> > > ### Author Response · Authors · 2024-08-13
> > >
> > > Thanks for your response. We acknowledge that the logical flow between the framework and the algorithm is weak in our current draft, which could lead to misunderstandings or make it difficult for readers to follow. To address this, we plan to reorganize the paper as follows:
> > >
> > > - Add the main intuition behind our theoretical framework which included in our global response at the beginning of Section 2 to make it easier for readers to understand and follow our framework.
> > > - Move 'Section 4.1 Motivation: An Idealized Method' to the end of Section 3, which mainly addresses the theoretical framework. This will create a clearer separation between the results on the theoretical framework and the real-world scenarios. We believe this change will help prevent any confusion by clearly distinguishing between the discussion of these two distinct settings.
> > > - At the beginning of Section 4, add an additional subsection to provide motivation for DASH by presenting the connection between DASH and the idealized method considered in the theoretical framework. This will help clarify how the theoretical findings relate to our practical proposal, DASH.
> > >
> > > We believe this reorganization will improve the paper.
> > >
> > > If you have further concerns or comments, we would be happy to hear them.

---

> ### Author Response · Authors · 2024-08-12
>
> Dear Reviewer t732,
>
> Thank you for your time and effort in reviewing our work. We greatly appreciate your valuable and constructive feedback. Given your busy schedule, we would be grateful if you could review our responses to ensure we've adequately addressed your concerns. If you have any further questions or suggestions, please let us know, and we'll be happy to address them.
>
> Thank you again for your valuable contribution to our research.

---

### Official Review · Reviewer_LBum · 2024-07-16

**Soundness:** 3
**Presentation:** 2
**Contribution:** 3
**Rating:** 7
**Confidence:** 3

**Summary:**

This paper investigates the reasons why warm-starting a neural network by pre-training it on a subset of the full training set leads to suboptimal performance when compared to training it on the full dataset from scratch. In particular, it proposes an abstract combinatorial model of feature learning that according to the authors captures the essence of this phenomenon. In this model, warm-starting leads to overfitting on feature noise that hinders learning new features when fine-tuning on the full distribution when compared to cold-starting. The authors then show that there exist an idealized learning algorithm in this abstract setting that can perform as well as cold-starting by forgeting the memorized noise prior to fine-tuning. Based on this theoretical intuition, they then propose Direction-Aware SHrinking (DASH) a technique that shrinks the learning step per-neuron based on its alignment with previous warm-starting epochs. DASH seems to outperform other techniques in a synthetic warm-starting benchmark.

**Strengths:**

1. **Intuitive and insightful theoretical model**: Personally, I find the discrete abstract model of learning of this paper to be very insightful. Theorems 3.4 and Theorems 4.1 are good examples of theoretical propositions that can help cement our empirical observations in deep learning. This model may not provide a rigorous path towards mathematically "proving" the observed warm-starting phenomenon, but it is useful to reason about it.
2. **Convincing explanations of warm-starting**: The provided explanations for the observed behavior of deep models under warm starting and the evidence to support them are solid and convincing.
3. **Good performance of DASH**: DASH seems to perform strongly against prior work on the studied warm-starting benchmarks.

**Weaknesses:**

1. **Unclear details of DASH**: The description of DASH in section 4.5 is not clear and the algorithm listing does not fully explain the algorithm (see questions). This makes it hard to assess the complexity of implementing DASH and its practical relevance.
2. **Limited evaluation of DASH in online learning**: The authors argue that one of the main motivations of studying the decrease of performance under warm-starting is the fact that many networks are trained in an online setting with new data being collected sequentially. However, the only evaluation of DASH is a synthetic warm-starting benchmark that only tests a narrow and unrealistic setting. I am not against this particular benchmark, but I believe the paper would have benefited from further investigations of DASH in a more realistic online setting (including comparing it against strong baselines in that space).
3. **Reasons for improved performance of DASH over cold-starting are not explained**: As the authors highlight, DASH seems to outperform cold-starting in their benchmarks. This phenomenon, however, cannot be explained by their theoretical model and thus it is a weakness of the model itself.

**Questions:**

I would appreciate a deeper description of how DASH works:
- Is the running average of gradients computed at every step or every epoch?
- Is the shrinking performed after every step, every epoch, every experiment?
- Can DASH be combined with any off-the-shelf deep learning optimizer? If so, how?
- What is the memory complexity of DASH?

If these questions are addressed and the explanations are clear and made part of the final manuscript I would be open to increase my score.

**Limitations:**

The authors briefly mention mention some of the limitations of their work (such as the unexplained outperformance of DASH over cold-starting) or specificity of their results to the stationary setting.

---

> ### Author Rebuttal · Authors · 2024-08-07
>
> We would like to express our appreciation to the reviewer for your valuable and constructive comments.  In the following, we address the points raised by the reviewer.
>
> ## **W1, Questions. Detailed description of DASH**
>
> We acknowledge that our description of DASH may have caused some confusion regarding the algorithm. DASH is a shrinking method that can be applied whenever new data is provided. As a result, the running average gradient computation (Q1) and shrinking (Q2) are performed at the beginning of the “experiment” (i.e., when a new data chunk is added). We realize that this description was omitted from Section 4.2 and was only mentioned in the Experimental Details (Section 5.1, line 359-360).
>
> Q3: Since DASH is an initialization scheme performed independently of the optimizer, it can be combined with any optimizer, such as SGD, Adam, or SAM. Indeed, we have provided results using two different optimizers: SGD and SAM.
>
> Q4: DASH calculates the weighted average gradient in training data sequentially, resulting in a memory complexity  proportional to the batch size and model size, as only the gradient needs to be stored. In a scenario where gradients are computed one at a time, DASH's memory complexity would be equivalent to the model size. Please refer to our global response for more details, where we have included experimental results about both the memory requirements and computational overhead of DASH.
>
> We plan to include these detailed explanations in our next revised version.
>
> ## **W2. Evaluation in online learning**
>
> Thank you for suggesting an evaluation of DASH in an online learning setting. We believe this is a great suggestion, as it addresses situations where storing all data is not feasible (e.g., due to memory constraints) and helps us explore how our algorithm can be applied to a wider range of scenarios. However, we'd like to clarify that our current setting is indeed realistic. It reflects real-world scenarios where new data continuously accumulates, such as in financial markets or social media platforms. This type of data accumulation is also studied in the work of Ash & Adams (2020).
>
> While we are not entirely sure of the specific online learning setting you have in mind, we have conducted experiments with a setup where, instead of accumulating and storing data, the model learns from new data as it becomes available. For example, we divided CIFAR-10 into 50 chunks and performed experiments with ResNet-18, where each experiment was conducted on a single chunk of data before moving to the next. All other settings remained consistent with experiment details provided in Section 5.1.
>
> To address the issue of catastrophic forgetting, which is common in online learning, we configured our experiments to apply DASH at specific intervals (e.g., every 10, 15, or 20 experiments), rather than after each experiment. Since there is no previous data available, we set $\alpha = 1$. After the 40th experiment, when the model had sufficiently learned, we stopped applying the shrinking process. We conducted similar experiments with the S&P method for comparison. It's worth noting that this variant is feasible because both DASH and S&P are algorithms that focus on adjusting the initialization of the model.
>
> We tested DASH and S&P with intervals of 10, 15, and 20 epochs. For both methods, we explored shrinkage parameter ($\lambda$) of 0.05, 0.1, and 0.3. Notably, DASH's test accuracy consistently surpassed that of S&P across all hyperparameter configurations. In Figure 8 of our PDF, we plotted the results using the best hyperparameters for each method. These results demonstrate that DASH outperforms both the warm-starting baseline and the S&P method in terms of test accuracy. Based on these findings, we can conclude that DASH is suitable for online learning scenarios. If you could provide more details about the specific online learning setting you mentioned, as well as any additional baselines, we'd be happy to conduct further experiments to explore this area.
>
> ## **W3. Reasons for improved performance of DASH over cold-starting**
>
> The reason our theoretical framework does not explain DASH's superior performance in cold-start scenarios may be due to the binary nature (learn/not learn, memorize/not memorize) of our framework. However, we can provide an intuitive explanation based on our approach of retaining features and forgetting noise. When applying DASH in practical scenarios, even features previously considered "learned" can be further "improved" (i.e., model learns the feature with a greater strength) after noise is forgotten through shrinking, due to the continuous nature of real learning scenarios. We believe further improvement of feature has contributed to the performance improvement.
>
> Thanks for your time and consideration.
>
> Best regards,
>
> Authors
>
> ---
> Reference
> - Jordan Ash and Ryan P Adams. On warm-starting neural network training. In NeurIPS 2020

---

> ### Author Response · Authors · 2024-08-12
>
> Dear Reviewer LBum,
>
> Thank you for taking the time to review our work and for providing such insightful and constructive feedback. We understand that you may have a busy schedule, but we wanted to follow up to ensure that our responses have sufficiently addressed your concerns. If you have any further questions or comments, we would be glad to hear them.

---

> > ### Comment · Reviewer_LBum · 2024-08-12
> > **Thank you for your rebuttal**
> >
> > I thank the authors for answering my questions and providing a thorough response to all reviewers. After reading the other reviews and the authors's responses I have decided to increase my score to 7: Accept as the authors have clarified the main
> >  details that were unclear in the implementation of DASH and provided new insightful experiments in a different online learning scenario (I apologize for my confusion in my previous review and acknowledge that their previous setting was also realistic for a data accumulation setup). I would encourage the authors to improve the description of DASH in a potential camera-ready version in any case.

---

> > > ### Author Response · Authors · 2024-08-12
> > >
> > > Thank you for your feedback and for reconsidering the score. We are glad to hear that our response addressed your concerns. We would also be happy to hear if you have any additional thoughts or suggestions.
> > >
> > > Best regards,
> > >
> > > Authors

---

### Author Rebuttal · Authors · 2024-08-07

We express our gratitude for your time and valuable comments. We would like to address the concerns and confusion raised by multiple reviewers.

## **Main intuition on our theoretical framework**
We would like to provide a clearer explanation of how our theoretical framework reflects the intuitive process of learning from image data. Figure 1 in the attached file illustrates the main ideas, which we believe will help readers better understand our approach.

Our framework is designed to capture characteristics of image data, where the input contains information relevant to both the image labels (which we refer to as “feature”, e.g., a cow’s ears, eyes, tail, and mouth in Figure 1) and irrelevant to the labels (which we refer to as “noise” e.g., sky, grass). Our framework is based on the insights from Shen et al. (2022) and incorporates these into a discrete learning framework.

Our training process is based on the belief that features that appear more frequently in train data are easier to learn, as gradients often align more strongly with these features. Thus, our framework is designed to sequentially learn the most frequent features (e.g., a cow’s ears) first.

As the model sequentially learns these features (e.g. a cow’s ears, eyes, mouth, & tail), it accumulates sufficient information to correctly classify the image in Figure 1 as a cow. Once the model correctly classifies data using these features (which we call "well-classified"), the impact of these data points on the learning decreases, as loss gradients become smaller as the predictions become more confident. To capture this characteristic, our framework evaluates the frequency of features solely on data points that are not well-classified.

As training proceeds, the algorithm may arrive at a point where all remaining features in non-well-classified train data points are not frequent enough to be learned. In this case, the noise in each data point has a faster learning speed, which leads the model to memorize noise instead of features to achieve 100% training accuracy.

## **Connection between ideal method and DASH**
First of all, we want to emphasize that, while DASH is inspired by the analysis of an abstract framework, it is a practical algorithm designed for real-world.

We acknowledge that our draft falls short of elucidating how DASH captures the principle of the idealized method. Here, we provide a clearer explanation of how DASH implements these principles in practice. A key component of DASH is shrinking by cosine similarity. We will detail how this technique helps the model retain features and forget noise.

The negative loss gradient for each neuron represents the direction to minimize the loss. As we discussed, minimizing loss involves two strategies: learning features/memorizing noise. We can view the negative loss gradient as a combination of directions for learning features and memorizing noise.

When a new chunk of data is added, features from the old data are likely to reappear frequently in the new data because these features are relevant to the class of data. In contrast, noise from the old data, which is class-independent and data-specific, is less likely to appear in the new data. As a result, neurons that have learned relevant features from previous experiments will have a high cosine similarity with the negative gradient of the new data. Conversely, neurons that have memorized noise will show a low cosine similarity with the negative gradient. Thus, shrinking by cosine similarity with the negative gradient of new data captures the essence of the idealized method: retaining features and forgetting noise.

We further experimented to validate that DASH aligns with our intuition. We trained a 3-layer CNN on CIFAR10, varying the size of the training dataset. The model trained using more data seems to learn more features, evidenced by the rising trend in test accuracy shown by the dashed line in Figure 2 in our attached pdf when we plot the cosine similarity greater than 0.1. Also, we observe a trend where the cosine similarity between the negative gradient from test data and the learned weights increases as the training data size grows, which aligns with our intuition.

## **Computation and memory overhead**
We will address the reviewers' concerns about the computational and memory overhead of DASH. First, it's important to note that DASH is applied only when new data is introduced, not at every step or epoch. Since DASH calculates the gradient of the whole dataset just once, its memory complexity is proportional to the batch size and model size.
To clarify concerns about training time, when comparing experiments with the same number of epochs, you can think of DASH as adding approximately one extra epoch to the total training time. Similarly, the computational overhead of DASH is roughly equivalent to running one additional epoch.
We provide aggregate results for FLOPS, CPU/CUDA memory usage, and training time for the first two experiments on CIFAR-10 using ResNet-18. These measurements were obtained using the `torch.profile` library, and the values were summed across all operations.

**Total FLOPS, CPU/CUDA Memory**
- **Warm** FLOPS: 19 trillion, CPU: 7.4 GB, CUDA: 724 GB, Training Time: 37s
- **Cold** FLOPS: 25 trillion, CPU: 9.6 GB, CUDA: 1010 GB, Training Time: 48s
- **S&P** FLOPS: 21 trillion, CPU: 8.6 GB, CUDA: 828 GB, Training Time: 42s
- **DASH** FLOPS: 21 trillion, CPU: 8.3 GB, CUDA: 827 GB, Training Time: 41s

It is worth noting that S&P requires one additional epoch compared to DASH in the second experiment. Despite this difference, our analysis shows that the total computational/memory overheads of S&P and DASH are remarkably similar.

We plan to include these discussions in the next revised version of our paper.

We hope our response helps to resolve any concerns and confusion.

Best regards,
Authors

---
Reference
- Shen et al. Data augmentation as feature manipulation. In ICML 2022

---

### Decision · Program_Chairs · 2024-09-25

**Decision:**

Accept (poster)

**Comment:**

This paper introduces Direction-Aware SHrinking (DASH), a novel method to mitigate plasticity loss in warm-started neural networks under stationary data distributions (i.e., IID distributions). The approach is well-grounded and supported by experimental results, although these results are only on relatively small, low-resolution, image datasets.

**Strengths:**
- **Novel Contribution:** DASH presents a unique approach to addressing the warm start problem, grounded in a solid theoretical framework.
- **Experimental Validation:** The method shows promising results on benchmark datasets.

**Weaknesses:**
- **Clarity and Theory:** The connection between the theoretical model and the practical implementation of DASH lacks clarity, particularly regarding the role of cosine similarity.
- **Evaluation Scope:** The evaluation is limited to toy datasets, raising concerns about DASH's scalability and performance on larger datasets like ImageNet-1K.
-**Efficiency:** While efficiency is one of the paper's central focuses, it is unclear how efficient the method is. While the number of steps (gradient updates) is reported, timing experiments are not detailed (although some timing information was provided during the rebuttal). Moreover, they only study SGD and SAM, rather than optimizers often used for training deep neural networks in the current era (e.g., Adam, AdamW).
- **Applicability:** The broader applicability of DASH, especially in non-stationary settings like class incremental learning, remains unclear. The authors did provide information during the rebuttal period, showing that their method worked poorly for class incremental learning.

**Recommendation: Accept.** The authors are encouraged to incorporate reviewer feedback and include the issues regarding non-stationary distributions in the limitations section.